# Experiments from unfinished Registered Reports in the Reproducibility Project: Cancer Biology

Timothy M Errington[1]*, Alexandria Denis[1†], Anne B Allison[2], Renee Araiza[3], Pedro Aza-Blanc[4], Lynette R Bower[3], Jessica Campos[3], Heidi Chu[5], Sarah Denson[3], Cristine Donham[6], Kaitlyn Harr[7], Babette Haven[8], Elizabeth Iorns[9], Jennie Kwok[5], Elysia McDonald[10], Steven Pelech[11,12], Nicole Perfito[9‡], Amanda Pike[5], Darryl Sampey[13], Michael Settles[14], David A Scott[15], Vidhu Sharma[5], Todd Tolentino[3], Angela Trinh[5], Rachel Tsui[9§], Brandon Willis[3], Joshua Wood[3], Lisa Young[5]

[1]Center for Open Science, Charlottesville, United States; [2]Piedmont Virginia Community College, Charlottesville, United States; [3]University of California, Davis, Davis, United States; [4]Independent consultant, San Diego, United States; [5]Applied Biological Materials, Richmond, Canada; [6]University of Georgia, Athens, United States; [7]University of Virginia, Charlottesville, United States; [8]Absorption Systems, Exton, United States; [9]Science Exchange, Palo Alto, United States; [10]Drexel University College of Medicine, Philadelphia, United States; [11]Kinexus Bioinformatics, Vancouver, Canada; [12]University of British Columbia, Vancouver, United States; [13]BioFactura, Frederick, United States; [14]Independent consultant, Naples, United States; [15]Sanford Burnham Prebys Medical Discovery Institute, La Jolla, United States

*For correspondence:
tim@cos.io

Present address: [†]Fordham University School of Law, New York, United States; [‡]Rarebase, Palo Alto, United States; [§]Komodo Health, San Francisco, United States

**Abstract** As part of the Reproducibility Project: Cancer Biology, we published Registered Reports that described how we intended to replicate selected experiments from 29 high-impact preclinical cancer biology papers published between 2010 and 2012. Replication experiments were completed and Replication Studies reporting the results were submitted for 18 papers, of which 17 were accepted and published by *eLife* with the rejected paper posted as a preprint. Here, we report the status and outcomes obtained for the remaining 11 papers. Four papers initiated experimental work but were stopped without any experimental outcomes. Two papers resulted in incomplete outcomes due to unanticipated challenges when conducting the experiments. For the remaining five papers only some of the experiments were completed with the other experiments incomplete due to mundane technical or unanticipated methodological challenges. The experiments from these papers, along with the other experiments attempted as part of the Reproducibility Project: Cancer Biology, provides evidence about the challenges of repeating preclinical cancer biology experiments and the replicability of the completed experiments.

## Editor's evaluation

This article provides a succinct presentation of the remaining unfinished Registered Reports for the Reproducibility Project: Cancer Biology. The article will be useful for evaluating the success of the reproducibility project.

## Introduction

The Reproducibility Project: Cancer Biology (RP:CB) was a collaboration between the Center for Open Science and Science Exchange that sought to address concerns about reproducibility in scientific research by conducting replications of selected experiments from a number of high-profile papers in the field of cancer biology (*Errington et al., 2014*). For each of these papers, a Registered Report detailing the proposed experimental designs and protocols for the replications was peer reviewed and published prior to data collection. When all experiments from a Registered Report were completed with interpretable data they were submitted and published as a Replication Study. However, there were numerous replication attempts that were not fully completed for various reasons as efforts were balanced attempting to complete as many experiments as possible across multiple Registered Reports, which were starting, finishing, and navigating challenges simultaneously over the course of the project. The decision to stop any individual experiment was influenced partly by the mundane technical or unanticipated methodological challenges unique to that experiment, and partly by factors related to time and cost estimates across all the replications that were ongoing as part of the Reproducibility Project: Cancer Biology as a whole. This includes four Registered Reports (*Blum and LaBarge, 2014*; *Evans and Griner, 2015*; *Greenfield and Griner, 2014*; *Incardona et al., 2015*) where experimental work was initiated but ultimately stopped without results, one Registered Report (*Bhargava et al., 2016b*) where the results were submitted as a Replication Study and rejected (which has been posted as a preprint [*Pelech et al., 2021*]), two Registered Reports (*Haven et al., 2016*; *Raouf et al., 2015*) where experimental work began with preliminary outcomes although experiments were not completed, and five Registered Reports (*Bhargava et al., 2016a*; *Chroscinski et al., 2015*; *Richarson et al., 2016*; *Sharma et al., 2016a*; *Sharma et al., 2016b*) where some experiments were completed. The present paper reports the preliminary outcomes and completed experiments for those seven partially completed attempts.

## Partial replication: a chromatin-mediated reversible drug-tolerant state in cancer cell subpopulations

There has been an increase in using quantitative bioinformatic approaches to model and understand the evolution of drug resistance. For example, to inform more effective clinical outcomes, there is a growing understanding of how tumors, the microenvironment, the immune system, and the epigenome interact and evolve (*McCoach and Bivona, 2019*). This is complicated by the challenges persister cell populations pose to therapy, drawing parallels to similar challenges in fighting microbial infections, with new bioinformatic approaches to decipher the complexities of heterogeneous cell populations, with a focus on drug-tolerant cells in cancers needed (*Vallette et al., 2019*). As part of the Reproducibility Project: Cancer Biology, we published a Registered Report (*Haven et al., 2016*) that described how we intended to replicate selected experiments from the paper 'A chromatin-mediated reversible drug-tolerant state in cancer cell subpopulations' (*Sharma et al., 2010*).

We attempted to independently replicate experiments that tested the phenotype of a small subpopulation of 'drug-tolerant persister' cells (DTPs) that are generated when PC9 cells, a protein-tyrosine kinase inhibitor (TKI)-sensitive non-small cell lung cancer cell line, is exposed to high concentrations of TKIs. As described in Protocol 1 in the Registered Report (*Haven et al., 2016*), we first attempted to determine the growth characteristics of DTPs. When attempting to determine the percentage of erlotinib-treated PC9 cells that became DTPs we used the same concentration (2 µM) and timing (9 days with media and drug replaced every 3 days) as the original study. We found ~6% of the cells remained viable and continued to proliferate, compared to vehicle-treated cells (*Figure 1A, B*), which was much higher than the ~0.3 % reported in the original study (*Sharma et al., 2010*). We confirmed the PC9 cells used in the replication attempt had a deletion in exon 19, which confers epidermal growth factor receptor(EGFR) TKI sensitivity (*Arao et al., 2004*), and did not contain a T790M mutation, which has been shown to confer erlotinib resistance (*Kobayashi et al., 2005*; *Pao et al., 2005*). We also performed a survival assay (*Figure 1C*) and checked EGFR kinase activation (*Figure 1D*), which both required higher concentrations of erlotinib to achieve a half-maximal inhibitory concentration (survival: ~0.15 µM; EGFR: >1 µM) compared to what was reported in the original study (survival: ~0.0085 µM; EGFR: <0.01 µM) (*Sharma et al., 2010*). Finally, we also observed similar results with erlotinib hydrochloride (HCl) (*Figure 1—figure supplement 1*), which was used in a paper that reported DTPs from PC9 cells (*Ramirez et al., 2016*). These preliminary outcomes

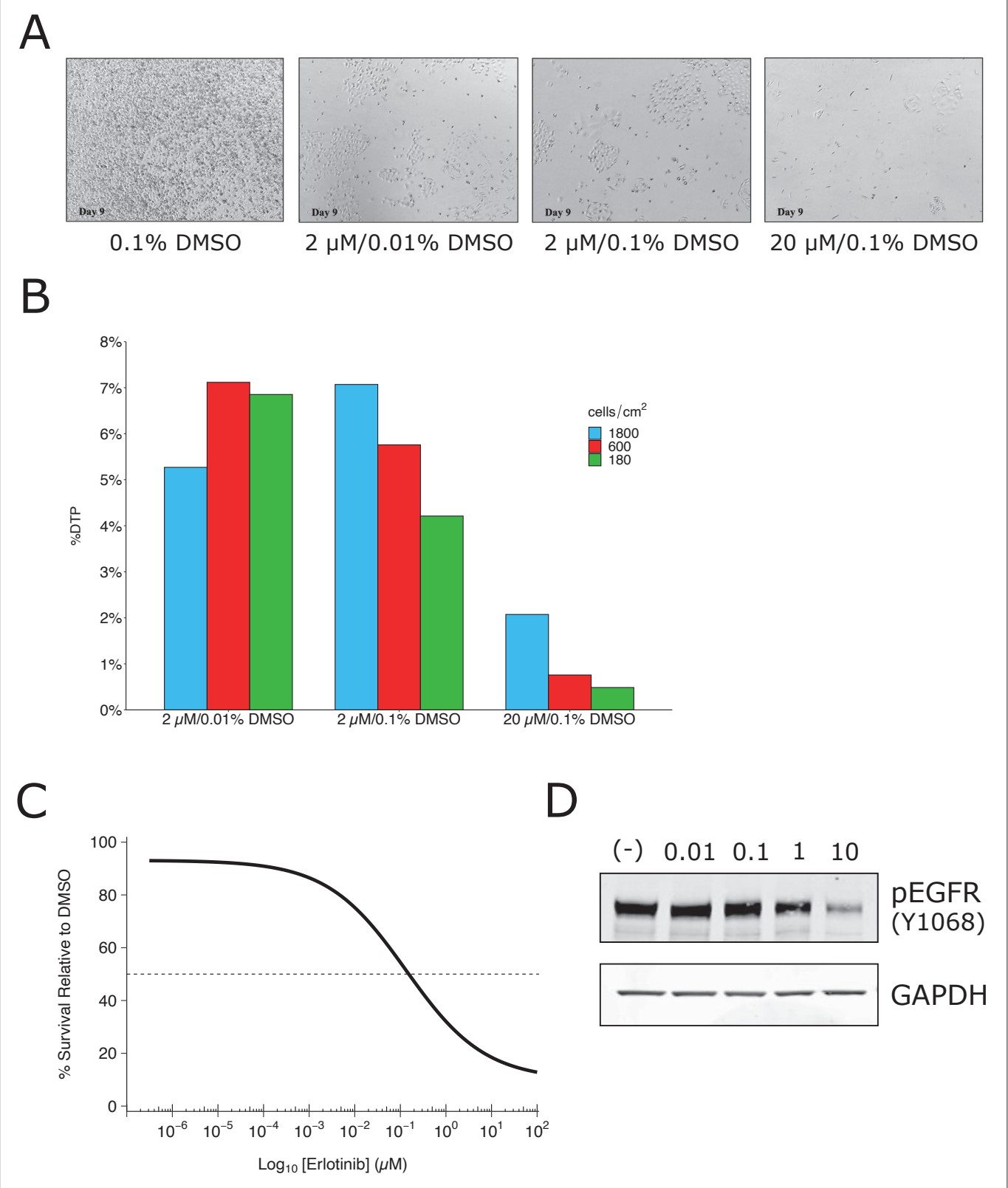

**Figure 1.** Replication attempt of *Sharma et al., 2010*. Growth characteristics of PC9 cells treated with various doses of erlotinib. (**A**) PC9 cells were plated at equal density and treated with dimethyl sulfoxide(DMSO) or the indicated dose of erlotinib (2 or 20 µM) for 9 days (fresh erlotinib was added every 3 days). Representative microscopic images of conditions on day 9. (**B**) Percent of drug-tolerant persisters (DTPs) as a percentage of DMSO control cells. PC9 cells were plated at various densities (cells/cm²) and treated with DMSO or the indicated concentrations of erlotinib. (**C**) Survival curve of

*Figure 1 continued on next page*

*Figure 1 continued*

PC9 cells treated with various doses of erlotinib for 72 hr. Percent survival is relative to DMSO-treated cells. The dashed line corresponds to 50% cell killing (absolute IC$_{50}$ = 0.15 µM). Data plotted from one independent biological repeat. (**D**) Representative Western blots of lysates from PC9 cells treated with increasing concentrations of erlotinib (0.01, 0.1, 1, and 10 µM) or DMSO (0.01%). Membranes were probed with phospho-EGFR (pEGFR)-specific antibodies. Glyceraldehyde 3-phosphate dehydrogenase (GAPDH) served as loading control. Relative expression of pEGFR (to DMSO) for each concentration (0.01, 0.1, 1, and 10 µM erlotinib) is: 1.1, 1.2, 0.8, and 0.2, respectively. Each experiment was performed independently twice. Additional details can be found at https://osf.io/xbign/.

The online version of this article includes the following figure supplement(s) for figure 1:

**Figure supplement 1.** Growth characteristics of PC9 cells treated with erlotinib hydrochloride (HCl).

indicated that while the PC9 cells retained their sensitivity to erlotinib, the exposure of the cells to erlotinib was not at a similar ~100 times the IC$_{50}$ concentration reported in the original study. This could be due to differences, such as variations of the cells used (e.g., PC9 cells used in this attempt were less resistant than those of the original study) or the potency of erlotinib (e.g., differing compound potency resulting from different stock solutions). To test a similar relative concentration we increased the concentration to 20 µM erlotinib and observed ~1 % of the cells remained viable after 9 days (*Figure 1A*). This indicates that DTPs were generated when PC9 cells were exposed to high concentrations of TKIs, similar to the original study; however, this raised questions for replicating the proposed experiments in the Registered Report. For example, how do the results differ when experiments are performed using the same dose as the original study (2 µM) compared to the higher concentration suggested in these preliminary results (20 µM), or is a different set of conditions necessary (e.g., is it necessary to achieve ~0.3% cell survival after 9 days?). We did not continue this experiment beyond this point and instead focused our efforts toward attempting to complete other replication experiments. Importantly, conducting experiments under different conditions could help provide insight into what conditions are necessary to obtain the original results. Interestingly, this was a point discussed during review of the Registered Report (*Haven et al., 2016*). To summarize, this replication attempt obtained preliminary outcomes although the experiments were not continued and thus these data do not address whether DTPs maintained viability through IGF-1 receptor signaling and an altered chromatin state dependent on the histone demethylase KDM5A.

## Partial replication: tumor vascularization via endothelial differentiation of glioblastoma stem-like cells

Of the handful of ways tumors acquire blood supply, transdifferentiation of tumor cells into blood vessel cells is a growing area of investigation. It has been reported that glioblastoma cells transdifferentiate into tumor endothelial cells, supporting the idea that glioblastoma stem cells contribute to the endothelial angiogenic properties of these tumors (*Ricci-Vitiani et al., 2010*; *Wang et al., 2010*). However, it has also been reported that glioblastoma stem cells transdifferentiate into pericytes rather than endothelial cells (*Cheng et al., 2013*). As part of the Reproducibility Project: Cancer Biology, we published a Registered Report (*Chroscinski et al., 2015*) that described how we intended to replicate selected experiments from the paper 'Tumour vascularization via endothelial differentiation of glioblastoma stem-like cells' (*Ricci-Vitiani et al., 2010*).

We independently replicated an experiment to determine the expression of the endothelial marker *Tie2* in patient-derived glioblastoma neurospheres compared to a glioblastoma cell line and an endothelial cell line. This is comparable to what was reported in Supplemental Figure 11C of *Ricci-Vitiani et al., 2010* and described in Protocol 1 in the Registered Report (*Chroscinski et al., 2015*). The replication used the same patient-derived glioblastoma neurospheres as the original study. We found that *Tie2* expression was low in the glioblastoma cells compared to higher expression in endothelial cells (*Figure 2A*), similar to the original study. A meta-analysis using a random-effects model was performed. The direction of the effects in the original study and this replication were in the same direction; however, the effect size point estimate of each study was not within the 95 % confidence interval (CI) of the other study, except for the comparison of the two glioblastoma cells (*Figure 2B*). The meta-analyses were not statistically significant. Importantly, the width of the CI for each study is a reflection of not only the confidence level (i.e., 95%), but also variability of the sample (e.g., *SD*) and sample size. To summarize, for this experiment we found results that were in the same direction as the original study and statistically significant where predicted.

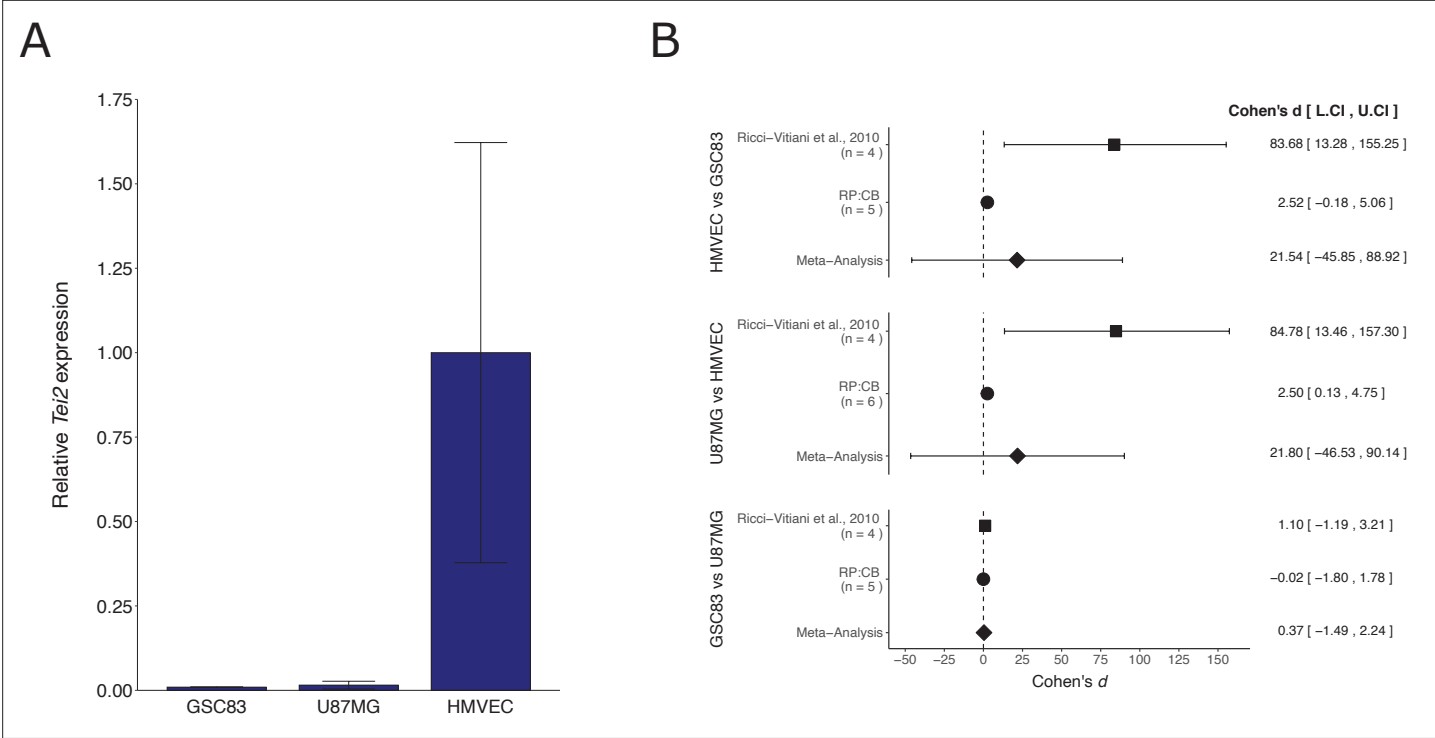

**Figure 2.** Replication attempt of *Ricci-Vitiani et al., 2010*. (**A**) *Tie2* expression in patient-derived glioblastoma neurospheres (GSC83), a glioblastoma cell line (U87MG), and an endothelial cell line (human dermal microvascular endothelial cells, HMVEC). Real-time quantitative reverse transcription PCR (qRT-PCR) was performed to detect *Tie2* and 18S rRNA expression. Relative expression (*Tie2*/18S rRNA) is presented for each cell type. Means reported and error bars represent *SD* from two (GSC83) or three (U87MG and HMVEC) independent biological repeats. One-way analysis of variance (ANOVA) of all groups: $F_{(2, 5)} = 5.90$, $p = 0.0484$. Planned contrasts between HMVEC and GSC83: $t_{(5)} = 2.78$, $p = 0.040$; U87MG and HMVEC: $t_{(5)} = 3.06$, $p = 0.028$; GSC83 and U87MG: $t_{(5)} = 0.017$, $p = 0.987$. (**B**) Meta-analysis of each effect. Effect size and 95 % confidence intervals are presented for *Ricci-Vitiani et al., 2010*, this replication study (Reproducibility Project: Cancer Biology, RP:CB), and a random-effects meta-analysis of those two effects. Cohen's *d* is a standardized difference between the two indicated measurements where a larger value indicates a difference in relative *Tie2* expression between the two cell types. Random-effects meta-analysis: HMVEC and GSC83: $p = 0.531$; U87MG and HMVEC: $p = 0.532$; GSC83 and U87MG: $p = 0.695$. Sample sizes used in *Ricci-Vitiani et al., 2010* and RP:CB are reported under the study name. Additional details can be found at https://osf.io/mpyvx/.

We also attempted to replicate an experiment that tested if glioblastoma stem-like cells (GSCs) that derive into endothelial cells contribute to tumor growth in vivo. As described in Protocol 2 in the Registered Report (*Chroscinski et al., 2015*) we first attempted to generate stable cells that expressed the herpes simplex virus thymidine kinase (*tk*) under the control of the transcription regulatory elements of *Tie2*, a vector conferring constitutive expression (i.e., *PGK*), or empty vector. A necessary requirement before proceeding with the xenograft experiment was achieving at least 80 % expression of the genes based on a GFP reporter. However, after multiple attempts, including obtaining new cells, plasmids, and incorporating changes to the protocol to improve infection efficiency and to enrich GFP expressing cells, we were unable to achieve this level for all cell populations. We did not continue this experiment beyond this point and instead focused our efforts toward attempting to complete other replication experiments. Importantly, this does not indicate the original result is unattainable, rather additional experimentation optimization is necessary. To summarize, this experiment was not completed and thus these data are unable to address whether selectively targeting endothelial cells generated by GSCs in a mouse xenograft model results in tumor reduction and degeneration.

### Partial replication: diverse somatic mutation patterns and pathway alterations in human cancers

As part of the Reproducibility Project: Cancer Biology, we published a Registered Report (*Sharma et al., 2016a*) that described how we intended to replicate selected experiments from the paper 'Diverse somatic mutation patterns and pathway alterations in human cancers' (*Kan et al., 2010*). Since then, *GNAO1*, which encodes the Gαo subunit of heterotrimeric guanine-binding proteins

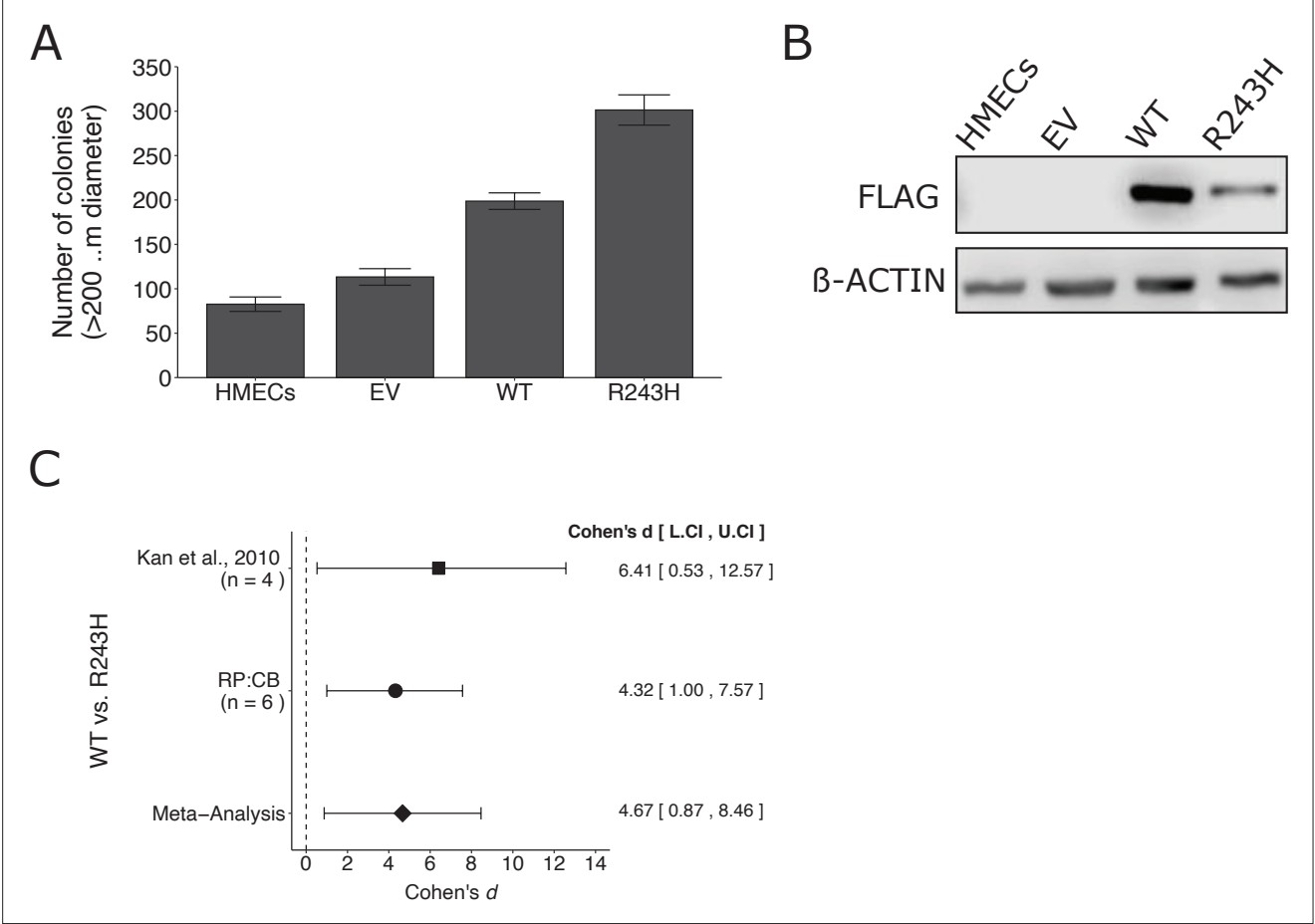

**Figure 3.** Replication of **Kan et al., 2010**. (**A**) Anchorage-independent growth of human mammary epithelial cells (HMECs). HMECs were left uninfected or stably expressing empty vector (EV), wild-type (WT) GNAO1, or GNAO1-R243H. Number of colonies formed after 3 weeks. PC9 cells treated with various doses of erlotinib. Means reported and error bars represent SEM from three independent biological repeats. Student's *t* test: *t*(4) = 5.29, *p* = 0.0061. (**B**) Representative Western blots of lysates from HMEC cells expressing the indicated conditions. Membranes were probed with FLAG-specific antibodies to detect FLAG-tagged GNAO1. B-ACTIN served as loading control. (**C**) Meta-analysis of each effect. Effect size and 95 % confidence intervals are presented for **Kan et al., 2010**, this replication study (Reproducibility Project: Cancer Biology, RP:CB), and a random-effects meta-analysis of those two effects. Cohen's *d* is a standardized difference between the two indicated measurements where a larger value indicates a difference in the number of colonies between the two conditions. Random-effects meta-analysis: *p* = 0.016. Sample sizes used in **Kan et al., 2010** and RP:CB are reported under the study name. Additional details can be found at https://osf.io/jpeqg/.

(G-proteins), has been identified as having a hypermethylated promoter region and being downregulated in colorectal cancer (**Hauptman et al., 2019**; **Hua et al., 2017**; **Yang et al., 2017**) and hepatoma carcinoma (**Xu et al., 2018**). Also, consistent with the understanding that decreased protein kinase activity leads to carcinogenesis, micro-RNAs (miR-141, miR-802, and miR-27A) have been found to inhibit *MAP2K4* in colon, tongue, and prostate carcinoma, respectively (**Ding et al., 2017**; **Wan et al., 2016**; **Wu et al., 2017**). Similarly, it has been reported that copy number loss of *MAP2K4* was observed in ductal carcinoma (**Pang et al., 2017**).

We independently replicated an experiment to test whether a somatic mutation in *GNAO1* promotes increased anchorage-independent growth compared to wild-type *GNAO1*. This is comparable to what was reported in Figure 3D–F of **Kan et al., 2010** and described in Protocols 1–3 in the Registered Report (**Sharma et al., 2016a**). Human mammary epithelial cells (HMECs) stably expressing wild-type *GNAO1* or *GNAO1^R243H* demonstrated increased colony formation in a soft agar assay compared to both vector control and uninfected HMECs (**Figure 3A, B**), similar to the original study that reported 1.8 times as many colonies in *GNAO1^R243H* expressing cells compared to wild-type *GNAO1* (**Kan et al., 2010**). A meta-analysis using a random-effects model was performed (**Figure 3C**). The two studies were consistent in direction and when considering if the effect size point

estimate of each study was within the 95% CI of the other study and the meta-analysis was also statistically significant, suggesting the R243H mutation promotes an increase in anchorage independent growth compared to cells expressing wild-type GNAO1. To summarize, we found results that were in the same direction as the original study and statistically significant where predicted.

We also attempted to replicate experiments that tested whether various somatic mutations in *MAP2K4*, a component of a kinase cascade that activates downstream MAP kinases, led to increased anchorage-independent growth and whether the mutations impair kinase activity in an in vitro kinase assay. This was described in Protocols 4 and 5 in the Registered Report (*Sharma et al., 2016a*). However, the soft agar assay was uninterpretable most likely due to a technical issue of the cells growing on the dish surface under the agar. We did not continue this experiment beyond this point and instead focused our efforts toward attempting to complete other replication experiments. Importantly, this does not indicate the original result is unattainable, rather additional experimentation optimization is necessary. To summarize, this experiment was not completed and thus these data are unable to address whether stable expression of *MAP2K4* somatic mutations in NIH3T3 cells results in increased anchorage independent growth and reduced kinase activity.

## Partial replication: kinase-dead BRAF and oncogenic RAS cooperate to drive tumor progression through CRAF

The Ras–Raf–MEK–ERK signal transduction cascade is one of the most common signaling pathways in human cancers. It has been reported that some RAF inhibitors paradoxically activate this pathway in the context of wild-type BRAF (*Halaban et al., 2010*; *Hatzivassiliou et al., 2010*; *Heidorn et al., 2010*; *Holderfield et al., 2013*; *Joseph et al., 2010*; *Lavoie et al., 2013*; *Poulikakos et al., 2010*). Consistent with this paradoxical activation, a kinase-dead BRAF was found to induce lung adenocarcinoma (*Nieto et al., 2017*). Similarly, kinase-impaired or kinase-dead (class 3) BRAF mutants were shown to be more dependent on CRAF for pathway activation and were susceptible to inhibition of activated Ras (*Yao et al., 2017*). Studies also reported effective inhibition of this pathway using pan-RAF inhibitors (*Vakana et al., 2017*) or a combination of RAF and MEK inhibitors (*Johannessen et al., 2010*; *Merchant et al., 2017*; *Molnár et al., 2018*; *Whittaker et al., 2015*). As part of the Reproducibility Project: Cancer Biology, we published a Registered Report (*Bhargava et al., 2016a*) that described how we intended to replicate selected experiments from the paper 'Kinase-dead BRAF and oncogenic RAS cooperate to drive tumor progression through CRAF' (*Heidorn et al., 2010*).

We independently replicated an experiment to test the paradoxical activation of MEK/ERK in wild-type BRAF and activated (mutant) RAS by BRAF inhibitors. This is comparable to what was reported in Figure 1A of *Heidorn et al., 2010* and described in Protocol 1 in the Registered Report (*Bhargava et al., 2016a*). We found A375 cells (mutant BRAF; wild-type NRAS) treated with BRAF inhibitors SB590885 and sorafenib blocked ERK activity, as did the MEK inhibitor PD184352, compared to controls; however, with D04 cells (wild-type BRAF; mutant NRAS) sorafenib and PD184352 blocked ERK activity while SB590885 did not (*Figure 4*). This is similar to the original study that reported all drugs blocked ERK activity in A375 cells, while there was increased ERK activity in D04 cells treated with 885A, a close analog of SB590885, and inhibited ERK in the other conditions (*Heidorn et al., 2010*). To summarize, we found results that were in the same direction as the original study and statistically significant, except for the effect of D04 cells treated with SB590885, which was not statistically significant.

We also attempted to replicate experiments testing the mechanism of the paradoxical activation through depletion of NRAS or CRAF or immunoprecipitation of ectopically expressed wild-type or mutant CRAF or BRAF. However, we did not continue this experiment beyond an initial attempt that was unsuccessful at consistently expressing the constructs as described in the Registered Report, thus requiring further optimization, and instead focused our efforts toward attempting to complete other replication experiments.

## Partial replication: COT drives resistance to RAF inhibition through MAP kinase pathway reactivation

As part of the Reproducibility Project: Cancer Biology, we published a Registered Report (*Sharma et al., 2016b*) that described how we intended to replicate selected experiments from the paper 'COT drives resistance to RAF inhibition through MAP kinase pathway reactivation' (*Johannessen*

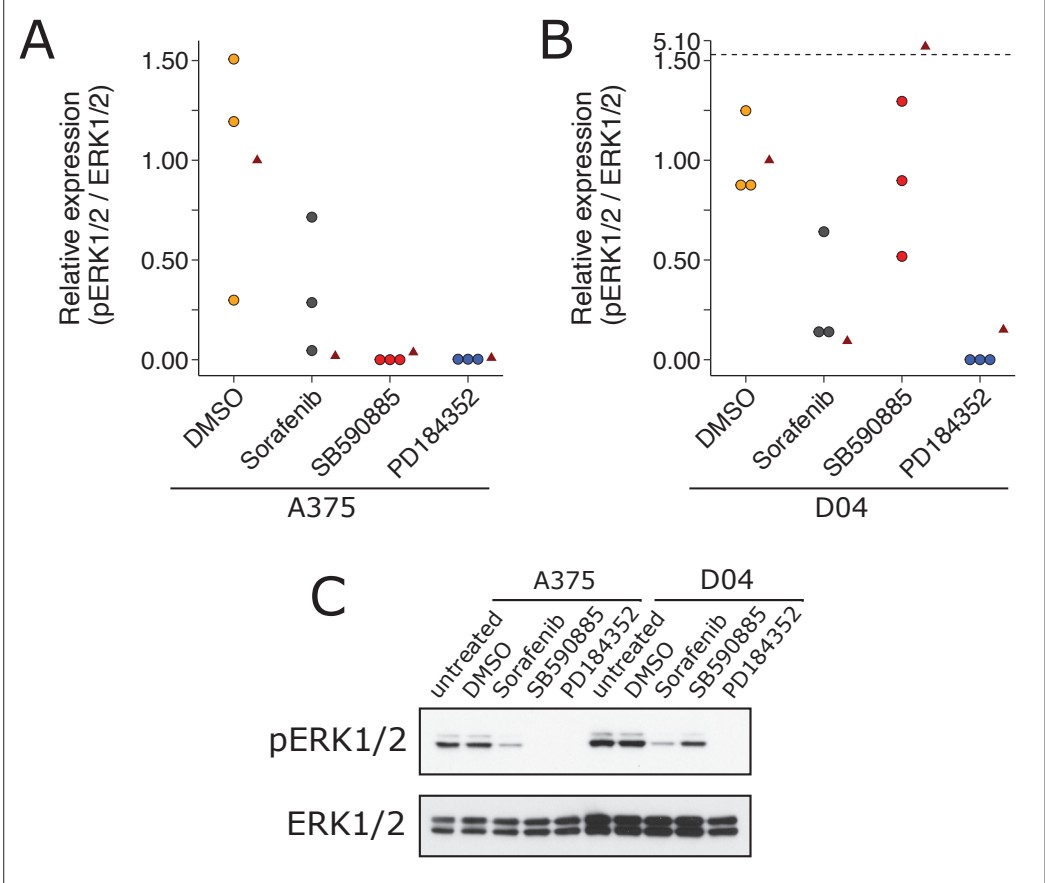

**Figure 4.** Replication attempt of *Heidorn et al., 2010*. ERK activation following treatment with BRAF inhibitors in A375 cells (mutant BRAF; wild-type NRAS) and D04 cells (wild-type BRAF; mutant NRAS). (**A**) A375 cells were treated with DMSO, sorafenib (10 µM), SB590885 (1 µM), or PD184352 (1 µM), or left untreated. Cells were harvested 4 hr later for Western blot analysis. Relative pERK1/2 expression are presented for each condition. Western blot bands were quantified, pERK1/2 was normalized to total ERK1/2, then for each biological repeat value was normalized to the untreated condition with expression presented relative to DMSO. Dot plot of independent biological repeats (n = 3). Data reported in **Figure 1A** of *Heidorn et al., 2010* displayed as a single point (red triangle) for comparison. Planned comparison (two-tailed Wilcoxon–Mann–Whitney test) between DMSO and all other conditions: $U = 2.40$, uncorrected $p = 0.0091$, Bonferroni corrected $p = 0.027$, Cliff's delta = 0.93, 95% confidence interval (CI) [0.63, 0.99]. (**B**) D04 cells were treated like in **A** and presented in the same way. Graph is separated at a dashed line to accommodate a value higher than the others. Data reported in **Figure 1A** of *Heidorn et al., 2010* displayed as a single point (red triangle) for comparison. Planned comparisons (two-tailed Wilcoxon–Mann–Whitney tests) between DMSO and sorafenib and PD184352: $U = 2.36$, uncorrected $p = 0.024$, Bonferroni corrected $p = 0.071$, Cliff's delta = 1.00, 95% CI [0.75, 1.00]; between DMSO and SB590885: $U = 1.96$, uncorrected $p = 0.10$, Bonferroni corrected $p = 0.30$, Cliff's delta = 0.11, 95% CI [−0.71, 80]. (**C**) Representative Western blots probed with pERK1/2 (T202/Y204)-specific antibodies. Total ERK1/2 served as loading control. Additional details can be found at https://osf.io/b1aw6/.

*et al., 2010*). Since then it has been reported that while targeting multiple components of the RAF–MEK–ERK pathway yield initial tumor regression in most patients with BRAF(V600E) mutation, resistance often still occurs (*Chatterjee and Bivona, 2019*; *Yu et al., 2019*). Recent literature highlights the success of combining druggable targets with immunotherapy (*Yu et al., 2019*).

We independently replicated an experiment to test if BRAF(V600E) cell lines expressing elevated levels of MAP3K8 are resistant to cellular growth inhibition by the RAF inhibitor PLX4720. This is comparable to what was reported in Figure 3D of *Johannessen et al., 2010* and described in Protocols 2 and 3 in the Registered Report (*Sharma et al., 2016b*). We intended to use OUMS-23 colon cancer cells, but were unable to propagate the cells despite troubleshooting with the supplier, so we switched to HT-29 colon cancer cells, which have high expression of *MAP3K8* (*Rouillard et al.,*

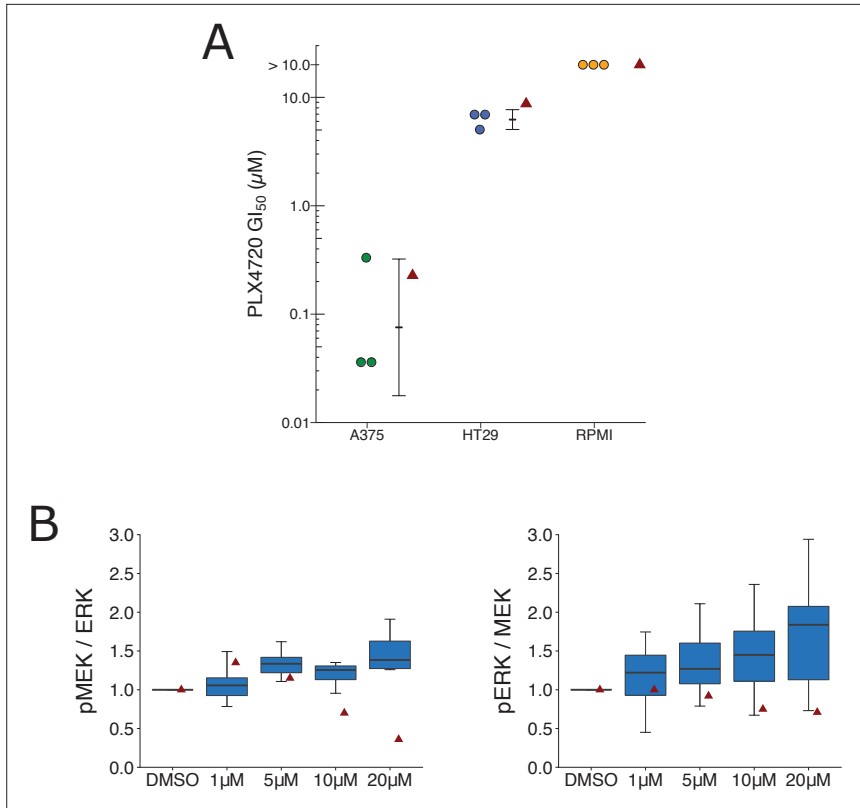

**Figure 5.** Replication attempt of *Johannessen et al., 2010*. (**A**) Cellular dose–response curves for RAF inhibitor (PLX4720) in BRAF(V600E) cell lines, A375, HT29, and RPMI-7951. Absolute half-maximum growth inhibition (GI$_{50}$) values (µM) were determined for each biological repeat. GI$_{50}$ values unable to be accurately estimated are reported as either >10 µM, which was the highest dose tested. Data reported in **Figure 3D** of *Johannessen et al., 2010* displayed as a single point (red triangle) for comparison. Where possible the mean and 95 % confidence interval (CI) of the replication data are shown. Planned comparison (Student's *t*-test) between A375 and HT29 GI$_{50}$ values: *t*(4) = 5.90, *p* = 0.0041; Cohen's *d* = 4.82, 95% CI [1.21, 8.36]. (**B**) Quantification of Western blots of lysates from RPMI-7951 cells treated with DMSO or the indicated doses of MAP3K8 kinase inhibitor for 1 hr. Membranes were probed with phospho-MEK- (pMEK), total ERK- (ERK), phospho-ERK- (pERK), or total MEK (MEK)-specific antibodies. pMEK levels were normalized to ERK, and pERK levels were normalized to MEK, and then to DMSO for each biological repeat [n = 8]. Note: Normalization of pMEK to ERK and pERK to MEK was a recommendation made by reviewers of the Registered Report (*Sharma et al., 2016b*). Box and whisker plots with median represented as the line through the box and whiskers representing values within 1.5 interquartile range (IQR) of the first and third quartiles. Data reported in **Figure 3I** of *Johannessen et al., 2010* is displayed as a single point (red triangle) for comparison. One-way analysis of variance (ANOVA) on pMEK/ERK data: *F*(3,28) = 1.93, uncorrected *p* = 0.147, Bonferroni corrected *p* = 0.295; pERK/MEK data: *F*(3,28) = 1.04, uncorrected *p* = 0.392, Bonferroni corrected *p* = 0.784. Planned one-sample *t*-test between 20 µM and a constant of 1 (DMSO-treated cells) on pMEK/ERK data: *t*(7) = 2.83, uncorrected *p* = 0.025, Bonferroni corrected *p* = 0.051; Cohen's *d* = −1.00, 95% CI [−1.84, −0.12]; pERK/MEK data: *t*(7) = 2.65, uncorrected *p* = 0.033, Bonferroni corrected *p* = 0.066; Cohen's *d* = −0.94, 95% CI [−1.76, −0.07]. Additional details can be found at https://osf.io/lmhjg/.

The online version of this article includes the following figure supplement(s) for figure 5:

**Figure supplement 1.** Cellular dose–response curves for each biological repeat and representative Western blot image of ERK and MEK phosphorylation.

*2016*; Johannessen, personal communication). We intended to confirm MAP3K8 protein expression in the cell lines, but were unable to use the same antibody used in the original study, which was discontinued from Santa Cruz Biotechnology, and we observed many nonspecific bands when two different anti-MAP3K8 antibodies were tested (see methods). We found while A375 cells exhibited a robust response to PLX4720, the two cell lines with high *MAP3K8* expression, RPMI-7951 and HT-29 resulted in half-maximal growth inhibition (GI$_{50}$) values of 5 µM or higher (*Figure 5A*, *Figure 5—figure*

*supplement 1A*), similar to the original study. To summarize, we found results that were in the same direction as the original study and statistically significant where predicted.

To test if MAP3K8 contributes to MEK/ERK activation in BRAF(V600E) cells, we treated RPMI-7591 cells with a small molecule MAP3K8 kinase inhibitor. This is comparable to what was reported in Figure 3I of *Johannessen et al., 2010* and described in Protocol 4 in the Registered Report (*Sharma et al., 2016b*). We found a dose-dependent activation of MEK and ERK phosphorylation, while the original study reported a dose-dependent suppression of MEK and ERK phosphorylation (*Figure 5B*, *Figure 5—figure supplement 1B*). To summarize, we found results that were in the opposite direction as the original study and not statistically significant.

We also attempted to replicate experiments further integrating the impact of ectopically expressed MAP3K8 in A375 cells to test if combinatorial MAPK pathway inhibition can override resistance to single agents. However, due to resource constraints we suspended the continuation of this replication attempt after an initial attempt was unsuccessful at expressing the constructs as described in the Registered Report, thus requiring further optimization.

## Partial replication: senescence surveillance of premalignant hepatocytes limits liver cancer development

As part of the Reproducibility Project: Cancer Biology, we published a Registered Report (*Raouf et al., 2015*) that described how we intended to replicate selected experiments from the paper 'Senescence surveillance of pre-malignant hepatocytes limits liver cancer development' (*Kang et al., 2011*). Since then a study reported tumor-specific CD8 T cells arose independently of CD4 T cells, but needed Th1-polarized CD4 T cells to effectively suppress tumorigenesis supporting the crucial role CD4 T cells play in tumor suppression by immunosurveillance (*Knocke et al., 2016*).

We attempted to independently replicate experiments that tested the immune clearance of premalignant senescent hepatocytes and the dependence on $CD4^+$ T cells. As described in the Registered Report (*Raouf et al., 2015*), the mechanism of mimicking aberrant oncogene activation was through stably delivering intrahepatic expression of oncogenic $Nras^{G12V}$ via hydrodynamic injection. We used the same transposase and transposon plasmids as the original study and blindly injected mice with plasmids to stably express $Nras^{G12V}$ or $Nras^{G12V/D38A}$. Wild-type (C.B-17) mice or SCID/beige mice that lacked a functional adaptive immunity were then blindly assessed for expression of Nras, and the prosenescence markers p21 and p16 at 6 or 30 days postinjection (*Figure 6A–F*) or wild-type (BL/6) or $CD4^{-/-}$ mice were blinded assessed for Nras expression 12 days postinjection (*Figure 6G*). Unexpectedly, the Nras expression for both experiments were quite low, with many at, or near, zero percent. For comparison, the original study reported ~15 % Nras positive cells for C.B-17 and SCID/beige mice at 6 days (*Kang et al., 2011*). Simultaneously, while p21 expression was lower than reported in the original study, particularly at day 6, p16 expression was much higher and variable. Importantly, negative controls (e.g., isotype antibodies) were at low expression levels for all conditions. The extremely low Nras expression levels confound interpretation of these data since it is unclear if there was unsuccessful expression (e.g., low transposition levels despite using the Sleeping Beauty transposon system and the hydrodynamics-based procedure that are highly effective methods [*Aronovich et al., 2011*]), the need to optimize the immunohistochemistry protocol (e.g., to address weak or absent staining [*Kim et al., 2016*]), or the expression had not occurred at detectable levels or were already cleared despite following the same timing as the original study (*Kang et al., 2011*). We did not continue this experiment beyond this point and instead focused our efforts toward attempting to complete other replication experiments. To summarize, this replication obtained preliminary outcomes although the experiments were not continued and thus these data are unable to address whether expression of oncogenic $Nras^{G12V}$ in mouse livers induced cellular senescence that were dependent on a functional adaptive immunity, specifically $CD4^+$ T cells.

## Partial replication: isocitrate dehydrogenase mutation impairs histone demethylation and results in a block to cell differentiation

Increases in methylation and blocks in differentiation caused by mutant isocitrate dehydrogenase (IDH) proteins have been described in human carcinogenesis (*Han et al., 2019*; *Waitkus et al., 2018*). As part of the Reproducibility Project: Cancer Biology, we published a Registered Report (*Richarson*

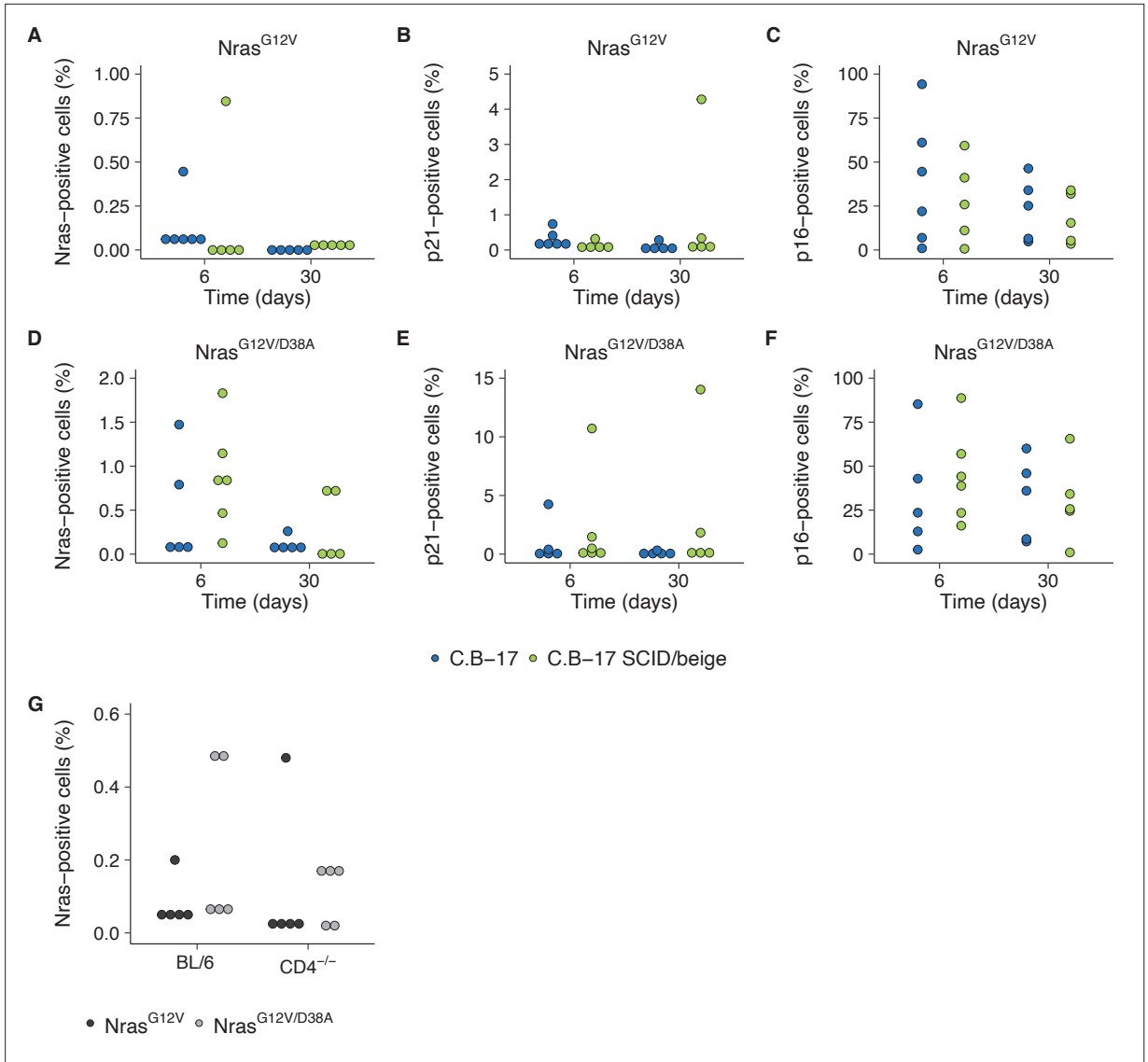

**Figure 6.** Replication attempt of *Kang et al., 2011*. (A–F) Quantification of Nras-, p21-, and p16-positive cells on liver sections from C.B-17 or C.B-17 SCID/beige mice six or 30 days after intrahepatic delivery of *Nras*[G12V] or *Nras*[G12V/D38A]. Percent positive cells were determined for each mouse. Five mice per group, except C.B-17 with *Nras*[G12V] at 6 days and C.B-17 SCID/beige with *Nras*[G12V/D38A] at 6 days, which had 6 mice per group. (G) Quantification of Nras-positive cells on liver sections from wild-type (BL/6) or CD4[−/−] mice 12 days after intrahepatic delivery of *Nras*[G12V] or *Nras*[G12V/D38A]. Percent positive cells were determined for each mouse (n = 5 per group). Additional details can be found at https://osf.io/82nfe/.

*et al., 2016*) that described how we intended to replicate selected experiments from the paper 'IDH mutation impairs histone demethylation and results in a block to cell differentiation' (*Lu et al., 2012*).

We independently replicated an experiment to test if expression of IDH mutations are associated with increased levels of various methylation markers and correlated with increased intracellular levels of the oncometabolite 2HG. HEK293T cells expressing ectopic wild-type IDH1, R132H mutant IDH1, wild-type IDH2, R172K mutant IDH2, or vector control were analyzed for accumulation of the metabolite 2HG by gas chromatography–mass spectrometry (GC–MS) and histone extracts were analyzed by Western blot for methylation status. This experiment is similar to what was reported in Figure 1B and Supplemental Figure 1 of *Lu et al., 2012* and described in Protocol 1 in the Registered Report (*Richarson et al., 2016*). We found 2HG levels in cells expressing mutant IDH to be ~200 times the levels detected in wild-type IDH (*Figure 7A*, *Figure 7—figure supplement 1B*). This compares to the original study, which reported an ~10–50 times increase in intracellular 2HG levels in cells expressing mutant compared to wild-type IDH (*Lu et al., 2012*). We also found a small, and not statistically

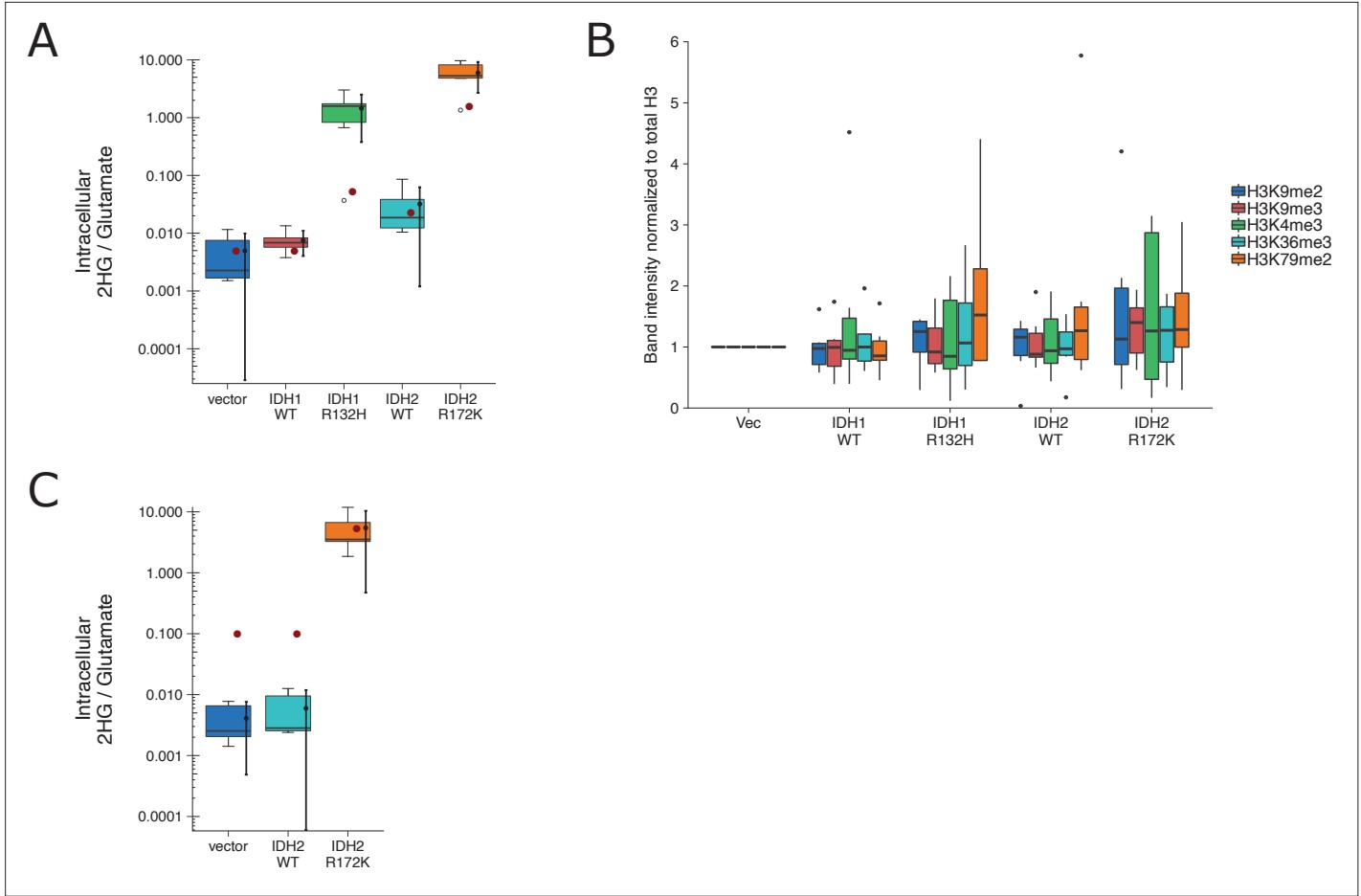

**Figure 7.** Replication attempt of *Lu et al., 2012*. (**A**) HEK293T cells transfected with wild-type (WT) or the indicated mutant of IDH1 or IDH2, or empty vector, were analyzed for intracellular metabolites 72 hr after transfection by gas chromatography–mass spectrometry (GC–MS). Quantitation of 2HG single intensity relative to glutamate was determined using total ion chromatograms (TIC) for each biological repeat (n = 6). Box and whisker plot with median represented as the line through the box and whiskers representing values within 1.5 interquartile range (IQR) of the first and third quartiles. Means as black dot and bold error bars represent 95% CI. Data estimated from the representative experiment reported in **Figure 1B** of *Lu et al., 2012* is displayed as a single point (red circle) for comparison. Statistical analysis was performed on $\log_{10}$-transformed data generated during this replication attempt. One-way analysis of variance (ANOVA) of IDH1 and IDH2 groups: $F(3, 20) = 56.8$, $p = 5.73 \times 10^{-10}$. Planned contrasts between IDH1 WT and IDH1 R132H: $t(20) = 8.33$, uncorrected $p = 6.23 \times 10^{-8}$, Bonferroni corrected $p = 1.25 \times 10^{-7}$, Cohen's $d = 4.81$, 95% CI [2.42, 7.15]; IDH2 WT and IDH2 R172K: $t(20) = 9.35$, uncorrected $p = 9.71 \times 10^{-9}$, Bonferroni corrected $p = 1.94 \times 10^{-8}$, Cohen's $d = 5.40$, 95% CI [2.79, 7.96]. (**B**) Western blot analysis for HEK293T cells transfected with WT or the indicated mutant of IDH1 or IDH2, or empty vector. Methylation status from histone extracts of the indicated markers were normalized to H3 and then to vector for each biological repeat (n = 6). Box and whisker plot with median represented as the line through the box and whiskers representing values within 1.5 IQR of the first and third quartiles. For each marker a one-way analysis of variance (ANOVA; or Kruskal–Wallis) was performed on IDH1 and IDH2 groups and the correlation with 2HG levels in A was performed. H3K4me3: $H(3) = 0.367$, uncorrected $p = 0.947$, Bonferroni corrected $p > 0.99$; correlation with 2HG: $t(12) = 1.12$, uncorrected $p = 0.275$, Bonferroni corrected $p = 0.99$. H3K9me2: $F(3,20) = 0.793$, uncorrected $p = 0.512$, Bonferroni corrected $p > 0.99$; correlation with 2HG: $t(12) = 1.53$, uncorrected $p = 0.140$, Bonferroni corrected $p = 0.699$. H3K9me3: $F(3,20) = 0.515$, uncorrected $p = 0.676$, Bonferroni corrected $p > 0.99$; correlation with 2HG: $t(12) = 1.35$, uncorrected $p = 0.191$, Bonferroni corrected $p = 0.957$. H3K36me3: $F(3,20) = 0.248$, uncorrected $p = 0.862$, Bonferroni corrected $p > 0.99$; correlation with 2HG: $t(12) = 0.725$, uncorrected $p = 0.476$, Bonferroni corrected $p > 0.99$. H3K79me2: $H(3) = 1.55$, uncorrected $p = 0.672$, Bonferroni corrected p $> 0.99$; correlation with 2HG: $t(12) = 0.752$, uncorrected $p = 0.460$, Bonferroni corrected $p > 0.99$. Planned contrasts for H3K9me2 between IDH1 WT and IDH1 R132H: $t(20) = -0.238$, uncorrected $p = 0.815$, Bonferroni corrected $p > 0.99$; IDH2 WT and IDH2 R172K: $t(20) = 1.31$, uncorrected $p = 0.203$, Bonferroni corrected $p > 0.99$. (**C**) 3T3-L1 cells were transduced to express the indicated proteins and analyzed and presented in the same way as **A**. Number of independent biological repeats (n = 5). Data estimated from the representative experiment reported in **Figure 2A** of *Lu et al., 2012* is displayed as a single point (red circle) for comparison. Statistical analysis was performed on $\log_{10}$-transformed data generated during this replication attempt. Student's t-test of IDH2 WT and IDH2 R172K: $t(8) = 14.3$, $p = 5.49 \times 10^{-7}$, Cohen's $d = 9.06$, 95% CI [4.53, 13.57]. Additional details can be found at https://osf.io/vfsbo/.

The online version of this article includes the following figure supplement(s) for figure 7:

**Figure supplement 1.** Replication attempt of *Lu et al., 2012*.

**Figure supplement 2.** Meta-analyses of effects from replication attempt of *Lu et al., 2012*.

significant, increase in histone methylation (*Figure 7B*, *Figure 7—figure supplement 1A*) that was not correlated with the observed 2HG levels. For comparison, the original study reported mutant IDH led to increased histone methylation compared to wild-type enzymes that were correlated with the intracellular 2HG levels (*Lu et al., 2012*). A meta-analysis using a random-effects model was performed for the histone methylation levels and the correlation with 2HG levels. The direction of the effects in the original study and this replication were in the same direction; however, the effect size point estimate of each study was not within the 95 % confidence interval (CI) of the other study, except for the comparison of H3K36me3 (*Figure 7—figure supplement 2*). The meta-analyses were not statistically significant. To summarize, we found results that were in the same direction as the original study and not statistically significant where predicted.

We also attempted to replicate an experiment to test if expression of mutant IDH2 in 3T3-L1 cells is associated with a block in differentiation to adipocytes. This is similar to what was reported in Figure 2A, B of *Lu et al., 2012* and described in Protocol 2 in the Registered Report (*Richarson et al., 2016*). We found 2HG levels in cells expressing mutant IDH2 to be ~1000 times the levels detected in wild-type IDH2 (*Figure 7C*, *Figure 7—figure supplement 2C, D*). This compares to the original study, which reported an ~50 times increase in intracellular 2HG levels in cells expressing mutant compared to wild-type IDH2 (*Lu et al., 2012*). The evaluation of differentiation, by Oil-Red-O staining and adipocyte marker expression, was uninterpretable though due to a lack of differentiation induced in control conditions, potentially due to the cells having lost their potential to differentiate (despite a successful pilot assay that resulted in differentiation by Oil-Red-O staining), or other methodological factors such as timing and staining procedure. We did not continue this experiment, or another experiment that tested the impact of the histone demethylase KDM4C on methylation and differentiation (Protocol 3 in the Registered Report), beyond this point and instead focused our efforts toward attempting to complete other replication experiments. To summarize, this replication attempt was not further continued and thus these data are unable to address whether mutant IDH2 is correlated with a block in differentiation and if this effect was mediated by the histone demethylase KDM4C.

## Conclusion

The details of attempting to replicate the experiments described above were combined with the rest of the experiments attempted as part of the Reproducibility Project: Cancer Biology to summarize the process of conducting this project and the challenges confronted (*Errington et al., 2021a*). Similarly, the outcomes from the completed experiments were aggregated with those from other replications from the project to examine the replicability of cancer biology research (*Errington et al., 2021b*). Not all replication experiments attempted were completed as resources were balanced while attempting to complete as many replication experiments as possible as part of the Reproducibility Project: Cancer Biology. A decision to stop an experiment does not necessarily indicate the original results are not replicable, as it is possible that devoting more project resources to any one experiment would have ultimately resolved the challenges. This includes overcoming mundane technical challenges such as optimization of techniques (e.g., transfection conditions) or overcoming unanticipated methodological challenges such as changing the protocol or reagents to account for an experimental model system producing unexpected outcomes (e.g., low expression level of an exogenous protein). Likewise, a decision to end an experiment does not indicate the original finding is replicable as it is possible that the challenges indicate an underappreciation of the conditions necessary for repeating the finding. While not all of the replication experiments attempted were completed, and thus are unable to provide outcomes to compare with the original published findings, sharing these incomplete experiments might provide useful information, such as preventing duplication by enabling others to build on these attempts and to extract the implicit information entwined with the observations. Sharing these outcomes also provides transparency of what was attempted and observed for the experiments described in the Registered Reports that were not fully completed and submitted as a Replication Study. Additionally, while some experiments were uninterpretable, because the outcomes were unexpected due to methodological challenges, it is all too easy to conflate an experiment as 'not working' when unexpected or null results are observed. This can be addressed by predefining the outcome-independent criteria for whether an experiment should be included, or excluded, such as in a preregistration (*Neves and Amaral, 2020*). However, this requires that all outcomes are shared, which is less common for null or incomplete findings (*Franco et al., 2014*). The Registered Reports publication

format is one way to accomplish this, as are a growing number of publication and reporting platforms (e.g., preprints) enabling researchers to disseminate their research outcomes regardless of whether it was an exciting finding or a perplexing attempt (*Bespalov et al., 2019*; *Sarabipour et al., 2019*). Much can be learned by openly sharing not only what we found, but what we tried along the way.

# Materials and methods

**Key resources table**

| Reagent type (species) or resource | Designation | Source or reference | Identifiers | Additional information |
|---|---|---|---|---|
| Cell line (*Homo sapiens*, male) | PC9 | Sigma-Aldrich | cat# 90071810; RRID:CVCL_B260 | lot# 14A030 |
| Chemical compound, drug | erlotinib | Cayman Chemical | cat# 10,483 | lot# 0459700-31 |
| Chemical compound, drug | erlotinib-HCl | LC Laboratories | cat# E-4007 | lot# BBE-108 |
| Software, algorithm | Cellometer Auto T4 Software | Nexcelom Bioscience | RRID:SCR_021656 | Version 3.1.1 |
| Software, algorithm | Image Studio Software | LI-COR Biosciences | RRID:SCR_015795 | |
| Antibody | rabbit-anti-phospho-EGFR (Y1068) | Cell Signaling Technology | cat# 3777; clone D7A5; RRID:AB_2096270 | 1:1000 dilution |
| Antibody | mouse anti-GAPDH | Thermo Fisher Scientific | cat# MA5-15738; clone GA1R; RRID:AB_10977387 | 1:2000 dilution |
| Antibody | IRDye 680RD-conjugated donkey anti-mouse | LI-COR Biosciences | cat# 926-68072; RRID:AB_10953628 | 1:15,000 dilution |
| Antibody | IRDye 800CW-conjugated donkey anti-rabbit | LI-COR Biosciences | cat# 926-32213; RRID:AB_621848 | 1:15,000 dilution |
| Software, algorithm | 2,100 Bioanalyzer Software | Agilent Technologies | RRID:SCR_019389 | Version 1.03 |
| Cell line (*Homo sapiens*, male) | U87MG | ATCC | cat#HTB-14; RRID:CVCL_0022 | |
| Cell line (*Homo sapiens*, male) | HMVEC | Lonza | CC-2516 | |
| Cell line (*Homo sapiens*, male) | GSC83 | doi:10.1038/nature09557 | RRID:CVCL_A9TL | Shared by Ricci-Vitiani lab, Istituto Superiore di Sanità |
| Cell line (*Homo sapiens*, female) | HMEC | Applied Biological Materials | cat# T0454; RRID:CVCL_B0CI | lot# RZ824921; infected with SV40 large T and small T antigen |
| Recombinant DNA reagent | pRetroX-IRES-ZsGreen1-empty vector | Clontech | cat# 632,520 | |
| Recombinant DNA reagent | pRetroX-FLAG-GNAO1$^{WT}$-IRES-ZsGreen1 | This paper | | |
| Recombinant DNA reagent | pRetroX-FLAG-GNAO1$^{R243H}$-IRES-ZsGreen1 | This paper | | |
| Cell line (*Homo sapiens*, female) | Phoenix | ATCC | cat# CRL-3213; RRID:CVCL_H716 | |
| Antibody | mouse anti-FLAG | Sigma-Aldrich | cat# F1804; clone M2; RRID:AB_262044 | 1:500 dilution |
| Antibody | mouse anti-β-ACTIN | Sigma-Aldrich | cat# A5441; clone AC-15; RRID:AB_476744 | 1:1000 dilution |

*Continued on next page*

*Continued*

| Reagent type (species) or resource | Designation | Source or reference | Identifiers | Additional information |
|---|---|---|---|---|
| Antibody | HRP-conjugated rabbit anti-mouse | Abcam | cat# ab6728; RRID:AB_955440 | 1:10,000 dilution |
| Software, algorithm | FACSDiva | BD Biosciences | RRID:SCR_016722 | Version 6.1.2 |
| Software, algorithm | ImageJ | doi:10.1038/nmeth.2089 | RRID:SCR_003070 | Version 1.51p |
| Cell line (*Homo sapiens*, female) | D04 | ABN Cell Line Bank, QIMR Berghofer Medical Research Institute | RRID:CVCL_H604 | |
| Cell line (*Homo sapiens*, female) | A375 | ATCC | cat# CRL-1619; RRID:CVCL_0132 | |
| Chemical compound, drug | sorafenib | Selleckchem | cat# S7397 | |
| Chemical compound, drug | SB590885 | Selleckchem | cat# S2220 | |
| Chemical compound, drug | PD184352 | Selleckchem | cat# S1020 | |
| Antibody | mouse anti-phospho-ERK1/2 (T202/Y204) | Cell Signaling Technology | cat# 9106; clone E10; RRID:AB_331768 | 1:1000 dilution |
| Antibody | Rabbit anti-ERK1/2 | Cell Signaling Technology | cat# 9102; RRID:AB_330744 | 1:1000 dilution |
| Cell line (*Homo sapiens*, female) | RPMI-7951 | ATCC | cat# HTB-66; RRID:CVCL_1666 | |
| Cell line (*Homo sapiens*, male) | OUMS-23 | JCRB | cat# JCRB1022; RRID:CVCL_3088 | |
| Cell line (*Homo sapiens*, female) | HT-29 | ATCC | cat# HTB-38; RRID:CVCL_0320 | |
| Chemical compound, drug | PLX4720 | Selleckchem | cat# S1152 | |
| Chemical compound, drug | MAP3K8 kinase inhibitor | EMD Millipore | cat# 616,373 | |
| Antibody | rabbit anti-phospho MEK1/2 (S217/221) | Cell Signaling Technology | cat# 9154; clone 41G9; RRID:AB_2138017 | 1:1000 dilution |
| Antibody | mouse anti-ERK1/2 | Cell Signaling Technology | cat# 4696; clone L34F12; RRID:AB_390780 | 1:1000 dilution |
| Antibody | rabbit anti-MEK1/2 | Cell Signaling Technology | cat# 8727; clone D1A5; RRID:AB_10829473 | 1:1000 dilution |
| Antibody | rabbit anti-Vinculin | Sigma-Aldrich | cat# V4139; RRID:AB_262053 | 1:20,000 dilution |
| Antibody | HRP-conjugated goat anti-rabbit | Cell Signaling Technology | cat# 7074; RRID:AB_2099233 | 1:1000 dilution |
| Antibody | HRP-conjugated horse anti-mouse | Cell Signaling Technology | cat# 7076; RRID:AB_330924 | 1:1000 dilution |
| Strain, strain background (*M. musculus*, C.BKa-Igh$^b$/lcrCrl, female) | C.B-17 wild-type | Charles River | RRID:IMSR_CRL:251 | |

*Continued on next page*

*Continued*

| Reagent type (species) or resource | Designation | Source or reference | Identifiers | Additional information |
|---|---|---|---|---|
| Strain, strain background (*M. musculus*, CB17. Cg-*Prkdc*$^{scid}$*Lyst*$^{bg}$/Crl, female) | SCID/beige | Charles River | RRID:IMSR_CRL:250 | |
| Strain, strain background (*M. musculus*, C57BL/6 J, female) | BL/6 wild-type | The Jackson Laboratory | RRID:IMSR_JAX:000664 | |
| Strain, strain background (*M. musculus*, B6.129S2-*Cd4*$^{tm1Mak}$/J, female) | CD4$^{-/-}$ | The Jackson Laboratory | RRID:IMSR_JAX:002663 | |
| Recombinant DNA reagent | pPGK-SB13 | doi:10.1038/nature10599 | | Shared by Zender lab, University of Tuebingen |
| Recombinant DNA reagent | pT/Caggs*Nras*$^{G12V}$ | doi:10.1038/nature10599 | | Shared by Zender lab, University of Tuebingen |
| Recombinant DNA reagent | pT/Caggs*Nras*$^{G12V/D38A}$ | doi:10.1038/nature10599 | | Shared by Zender lab, University of Tuebingen |
| Antibody | mouse anti-p16 | Abcam | cat# ab54210; clone 2D9A12; RRID:AB_881819 | 1:50 dilution |
| Antibody | mouse anti-p21 | BD Biosciences | cat# 556431; clone SXM30; RRID:AB_396415 | 1:50 dilution |
| Antibody | mouse anti-Nras | Santa Cruz Biotechnology | cat# sc-31; clone F155; RRID:AB_628041 | 1:100 dilution |
| Antibody | mouse IgG1 isotype control | Sigma-Aldrich | cat# M5284; clone MOPC21; RRID:AB_1163685 | 1:50 dilution |
| Antibody | biotinylated goat anti-mouse | Thermo Fisher Scientific | ted goat anti-mouse (Thermo Fisher Scientific) | |
| Software, algorithm | CaseViewer | 3DHISTECH | RRID:SCR_017654 | Version 2.2 |
| Cell line (*Homo sapiens*, female) | HEK293T | ATCC | cat# CRL-3216; RRID:CVCL_0063 | |
| Cell line (*M. musculus*, male) | 3T3-L1 | ATCC | cat# CL-173; RRID:CVCL_0123 | |
| Recombinant DNA reagent | pLPC empty vector | This paper | | |
| Recombinant DNA reagent | pLPC-IDH1 wildtype | This paper | | |
| Recombinant DNA reagent | pLPC-IDH1$^{R132H}$ | This paper | | |
| Recombinant DNA reagent | pLPC-IDH2 wildtype | This paper | | |
| Recombinant DNA reagent | pLPC-IDH2$^{R172K}$ | This paper | | |
| Antibody | rabbit anti-H3 | Cell Signaling Technology, | cat# 4499; clone D1H2; RRID:AB_10544537 | 1:1000 dilution |

*Continued on next page*

*Continued*

| Reagent type (species) or resource | Designation | Source or reference | Identifiers | Additional information |
|---|---|---|---|---|
| Antibody | rabbit anti-H3K4me3 | Millipore | cat# 17–614; RRID:AB_11212770 | 1:2000 dilution |
| Antibody | rabbit anti-H3K36me3 | Abcam | cat# ab9050; RRID:AB_306966 | 1 µg/ml dilution |
| Antibody | rabbit anti-H3K79me2 | Cell Signaling Technology | cat# 9757; RRID:AB_2118448 | 1:1000 dilution |
| Antibody | rabbit anti-H3K9me2 | Cell Signaling Technology | cat# 9753; RRID:AB_659848 | 1:1000 dilution |
| Antibody | rabbit anti-H3K9me3 | Abcam | cat# ab8898; RRID:AB_306848 | 1 µg/ml dilution |
| Antibody | rabbit anti-IDH1 | Proteintech | cat# 12332-1-AP; RRID:AB_2123159 | 1:1500 dilution |
| Antibody | mouse anti-IDH2 | Abcam | cat# ab55271; clone 5F11; RRID:AB_943793 | 1 µg/ml dilution |
| Software, algorithm | R project for statistical computing | https://www.r-project.org | RRID:SCR_001905 | Version 4.1.0 |
| Software, algorithm | metafor | doi:10.18637/jss.v036.i03 | RRID:SCR_003450 | Version 3.0-2 |

## Partial replication: a chromatin-mediated reversible drug-tolerant state in cancer cell subpopulations

A detailed description of all protocols can be found in the Registered Report (*Haven et al., 2016*) with detailed protocol information for attempted experiments available on OSF (https://osf.io/xbign/). A summary of methodological details of results reported are described below.

PC9 cells (Sigma-Aldrich, cat# 90071810, RRID:CVCL_B260) were grown in RPMI supplemented with 4.5 g/l glucose, 5 % fetal bovine serum (FBS), and 1 % penicillin/streptomycin at 37°C in a humidified atmosphere at 5 % $CO_2$. Cells were confirmed to be free of mycoplasma contamination as well as confirmed to be the indicated cells by STR DNA profiling using tests performed by DDC Medical (Fairfield, OH). Genomic DNA from PC9 cells were also assessed for exon 19 deletion by PCR (forward primer: 5′-GGTAACATCCACCCAGATCAC-3′; reverse primer: 5′-CAGCTGCCAGACAT-GAGAAA-3′) and status of amino acid position 790 in exon 20 of the EGFR gene by PCR (forward primer: 5′-CCATGCGAAGCCACACTGA-3′; reverse primer: 5′-GTGAGGATCCTGGCTCCTT-3′) and analyzed with a 2100 Bioanalyzer (Agilent Technologies, cat# G2939A), software version 1.03.

To generate DTPs, cells were plated at various densities. The next day fresh medium with various concentrations of erlotinib (Cayman Chemical, cat# 10483) or erlotinib HCl (LC Laboratories, cat# E-4007). Media was replaced every 3 days. Nine days after flasks were imaged, cells were trypsinized and centrifuged down at 100 × *g* for 5 min before determining cell density and viability using a Cellometer Auto T4 (Nexcelom Bioscience) and Cellometer Auto T4 Software (Nexcelom Bioscience, RRID:SCR_021656), version 3.1.1.

Cell viability assays were performed after plating cells at 2500 cells/well in a 96-well plate and treating with various concentrations of erlotinib or erlotinib HCl. Three days later plates were assessed by Cell Titer Glo (Promega, cat# G7571) following the manufacturer's instructions and a Synergy two multimode plate reader (Bio-Tek) or Syto60 (Molecular Probes, cat# S11342) staining and an Odyssey CLx Infrared Imaging System and Image Studio Software (LI-COR Biosciences).

Western blot analysis was performed on lysate from cells treated with various doses of erlotinib for 2 hr. Cells were harvested in ice-cold radioimmunoprecipitation assay (RIPA) buffer (Sigma-Aldrich, cat# R0278) supplemented with 1× Complete Mini Protease Inhibitor Cocktail (Roche Diagnostics, cat# 11836153001), 1× Halt Phosphatase Inhibitor Cocktail (Thermo Scientific, cat# 1862495), and 1 mM PMSF (Sigma-Aldrich, cat# 93482). 25 µg of cell lysate was separated in a 4–12% Bis-Tris gel using 1× MOPS Running Buffer as described in the Registered Report. Gels were transferred to nitrocellulose, blocked for 1 hr at room temperature with 5% wt/vol nonfat dry milk in 1× Tris-buffered saline (TBS) with 0.1% Tween-20 (TBST). Membranes were probed overnight at 4°C with rabbit anti-phospho-EGFR

(Y1068) (Cell Signaling Technology, cat# 3777, clone D7A5, RRID:AB_2096270) at 1:1000 dilution or mouse anti-GAPDH (Thermo Fisher Scientific, cat# MA5-15738, clone GA1R, RRID:AB_10977387) at 1:2000 dilution in 1 % bovine serum albumin (BSA)/TBST. Following 3× 5 min washes with TBST, membranes were incubated with the appropriate secondary antibody for 1 hr at room temperature: IRDye 680RD-conjugated donkey anti-mouse (LI-COR, cat# 926-68072, RRID:AB_10953628) or IRDye 800CW-conjugated donkey anti-rabbit (LI-COR, cat# 926-32213, RRID:AB_621848) at 1:15,000 dilution in 1% wt/vol BSA/TBST. Membranes were scanned with an Odyssey CLx Infrared Imaging System and Image Studio Software (LI-COR Biosciences).

## Partial replication: tumor vascularization via endothelial differentiation of GSCs

A detailed description of all protocols can be found in the Registered Report (*Chroscinski et al., 2015*) with detailed protocol information for attempted experiments available on OSF (https://osf.io/mpyvx/). A summary of methodological details of results reported are described below.

U87MG cells (ATCC, cat#HTB-14, RRID:CVCL_0022) were grown in DMEM supplemented with 4.5 g/l glucose, 10 % FBS, and 1 % penicillin/streptomycin at 37°C in a humidified atmosphere at 5 % $CO_2$. HMVEC cells (Lonza, cat# CC-2516) were grown in endothelial growth medium-2 microvascular (Lonza, cat# CC-3202). GSC83 cells (RRID:CVCL_A9TL), which were used in the original study were shared by Dr. Lucia Ricci-Vitiani (Istituto Superiore di Sanità) and were grown in stem cell medium as specified in the Registered Report. All cells were confirmed to be free of mycoplasma contamination (Sigma-Aldrich, cat# MP0035) as well as confirmed to be the indicated cells by STR DNA profiling using tests performed by DDC Medical (Fairfield, OH). Total RNA was extracted from cells using TRI reagent and reverse transcribed to cDNA using a First-strand cDNA synthesis Kit (GE Healthcare, cat# GE27-9261-01) following the manufacturer's instructions. qPCR reactions were performed in technical triplicate with the TaqMan probes and cycling conditions described in the Registered Report using a 7,500 Fast Real-Time PCR system (Applied Biosystems, RRID:SCR_018051).

## Partial replication: diverse somatic mutation patterns and pathway alterations in human cancers

A detailed description of all protocols can be found in the Registered Report (*Sharma et al., 2016a*) with detailed protocol information for attempted experiments available on OSF (https://osf.io/jpeqg/). A summary of methodological details of results reported are described below.

HMEC cells (Applied Biological Materials, cat# T0454, RRID:CVCL_B0CI) are a hTERT-immortalized human mammary epithelial cell line that were infected with SV40 large T and small T antigen (Applied Biological Materials, cat# G258) and grown in PriGrow IV Medium (Applied Biological Materials, cat# TM004) supplemented with 12.5 ng EGF (Cell Biologics, cat# Z100135), 50 µM ascorbic acid (Sigma-Aldrich, cat# A92902), 2 nM estradiol (Sigma-Aldrich, cat# E2758), 1 µg/ml insulin (Sigma-Aldrich, cat# I9278), 2.8 µM hydrocortisone (Sigma-Aldrich, cat# H0888), 0.1 mM ethanolamine (Sigma-Aldrich, cat# E0135), 0.1 mM L-glutamine, 15 nM sodium selenite (Sigma-Aldrich, cat# S5261), 1 ng/ml cholera toxin (Sigma-Aldrich, cat# C9903), 1 % FBS, and 1 % penicillin/streptomycin at 37°C in a humidified atmosphere at 5 % $CO_2$. HMEC cells were infected with retroviruses generated from transfected Phoenix cells (ATCC, cat# CRL-3213, RRID:CVCL_H716) using the Retro-packaging Mix (Applied Biological Materials, cat# E-510) and pRetroX-IRES-ZsGreen1-empty vector (Clontech, cat# 632520), pRetroX-FLAG-GNAO1$^{WT}$-IRES-ZsGreen1, or pRetroX-FLAG-GNAO1$^{R243H}$-IRES-ZsGreen1. Retrovirus titration was performed using a qRT-PCR kit (Takara, cat# 631453) following the manufacturer's instructions to determine multiplicity of infection. Following transduction, cells were sorted by FACS and the top 10% were selected based on GFP fluorescence. Flow cytometry analysis was performed on a FACSAria II (BD Biosciences) and analyzed with FACSDiva software (BD Biosciences, RRID:SCR_016722), version 6.1.2. All cells were confirmed to be free of mycoplasma contamination as well as confirmed to be the indicated cells by STR DNA profiling using tests performed by DDC Medical (Fairfield, OH).

Western blot analysis was performed as described in the Registered Report and probed with mouse anti-FLAG (Sigma-Aldrich, cat# F1804, clone M2, RRID:AB_262044) at 1:500 dilution or mouse anti-β-ACTIN (Sigma-Aldrich cat# A5441, clone AC-15, RRID:AB_476744) at 1:1000 dilution. Followed by incubation with the appropriate secondary antibody: horseradish peroxidase (HRP)-conjugated rabbit

anti-mouse (abcam, cat# ab6728, RRID:AB_955440) at 1:10,000 dilution. Anchorage-independent colony formation assays were performed as described in the Registered Report using ImageJ software (RRID:SCR_003070), version 1.51 p (*Schneider et al., 2012*).

## Partial replication: kinase-dead BRAF and oncogenic RAS cooperate to drive tumor progression through CRAF

A detailed description of all protocols can be found in the Registered Report (*Bhargava et al., 2016a*) with detailed protocol information for attempted experiments available on OSF (https://osf.io/b1aw6/). A summary of methodological details of results reported are described below.

D04 cells (ABN Cell Line Bank, QIMR Berghofer Medical Research Institute, RRID:CVCL_H604) were grown in RPMI supplemented with 10 % FBS and 1 % penicillin/streptomycin at 37°C in a humidified atmosphere at 5 % $CO_2$. A375 cells (ATCC, cat# CRL-1619, RRID:CVCL_0132) were grown in DMEM supplemented with 10 % FBS and 1 % penicillin/streptomycin at 37°C in a humidified atmosphere at 5 % $CO_2$. Cells were confirmed to be free of mycoplasma contamination as well as confirmed to be the indicated cells by STR DNA profiling using tests performed by DDC Medical (Fairfield, OH).

A375 or D04 cells were treated either with DMSO, 10 μM sorafenib (Selleckchem, cat# S7397), 1 μM SB590885 (Selleckchem, cat# S2220), or 1 μM PD184352 (Selleckchem, cat# S1020) for 4 hr and then harvested for Western blot analysis. Western blot analysis was performed as described in the Registered Report and 7–12 μg of lysate were probed with mouse anti-phospho-ERK1/2 (T202/Y204) (Cell Signaling Technology, cat# 9106, clone E10, RRID:AB_331768) at 1:1000 dilution or rabbit anti-ERK1/2 (Cell Signaling Technology, cat# 9102, RRID:AB_330744) at 1:1000 dilution. Followed by incubation with the appropriate HRP-conjugated secondary antibody. Quantification was performed using ImageJ software (RRID:SCR_003070), version 1.51 p (*Schneider et al., 2012*).

## Partial replication: COT drives resistance to RAF inhibition through MAP kinase pathway reactivation

A detailed description of all protocols can be found in the Registered Report (*Sharma et al., 2016b*) with detailed protocol information for attempted experiments available on OSF (https://osf.io/lmhjg/). A summary of methodological details of results reported are described below.

RPMI-7951 cells (ATCC, cat# HTB-66, RRID:CVCL_1666) and OUMS-23 cells (JCRB, cat# JCRB1022, RRID:CVCL_3088) were grown in RPMI (ATCC, cat# 30-2001) supplemented with 10 % FBS and 1 % penicillin/streptomycin at 37°C in a humidified atmosphere at 5 % $CO_2$. A375 cells (ATCC, cat# CRL-1619, RRID:CVCL_0132) were grown in DMEM supplemented with 10 % FBS and 1 % penicillin/streptomycin at 37°C in a humidified atmosphere at 5% $CO_2$. HT-29 cells (ATCC, cat# HTB-38, RRID:CVCL_0320) were grown in McCoys 5 A (ATCC, cat# 30-2007) supplemented with 10 % FBS and 1 % penicillin/streptomycin at 37°C in a humidified atmosphere at 5 % $CO_2$. Cells were confirmed to be free of mycoplasma contamination as well as confirmed to be the indicated cells by STR DNA profiling using tests performed by DDC Medical (Fairfield, OH).

Cellular dose–response assays were performed with empirically determined seeding densities of each cell line by seeding at a range of densities outlined in the Registered Report. Cell viability was determined for the cells 5 days after seeding with the WST1 viability assay (Roche, cat# 11644807001) following the manufacturer's instructions. The number of cells seeded that remained subconfluent by 5 days with a signal still in the exponential phase at the end of the assay were used to test the inhibitory effects of compounds. RPMI-7951 cells were seeded at 3000 cells/well, A375 cells were seeded at 1500 cells/well, and HT-29 cells were seeded at 3200 cells/well in 96-well plates and incubated overnight. The following day cells were treated with serial dilutions of PLX4720 (Selleckchem, cat# S1152) with a final DMSO concentration of 0.1% vol/vol. Cells were incubated for 96 hr before cell viability was determined with the WST1 viability assay. All conditions were done with six technical replicates. For each biological repeat, the average background from media-only wells was subtracted from each well before values were normalized to the average DMSO wells. These data were fitted to a four-parameter curve for each biological repeat to calculate the absolute $GI_{50}$ values. $GI_{50}$ values unable to be accurately estimated following published guidelines (*Sebaugh, 2011*) were reported as the maximum dose tested.

RPMI-7951 cells were treated either with DMSO or various doses of MAP3K8 kinase inhibitor (EMD Millipore, cat# 616373) for 1 hr and then harvested for Western blot analysis as described in

the Registered Report. 20 μg of lysate was probed with mouse anti-phospho-ERK1/2 (T202/Y204) (Cell Signaling Technology, cat# 9106, clone E10, RRID:AB_331768) at 1:1000 dilution, rabbit anti-phospho MEK1/2 (S217/221) (Cell Signaling Technology, cat# 9154, clone 41G9, RRID:AB_2138017) at 1:1000 dilution, mouse anti-ERK1/2 (Cell Signaling Technology, cat# 4696, clone L34F12, RRID:AB_390780) at 1:1000 dilution, rabbit anti-MEK1/2 (Cell Signaling Technology, cat# 8727, clone D1A5, RRID:AB_10829473) at 1:1000 dilution, or rabbit anti-Vinculin (Sigma-Aldrich, cat# V4139, RRID:AB_262053) at 1:20,000 dilution. Followed by incubation with the appropriate secondary antibody: HRP-conjugated goat anti-rabbit (Cell Signaling Technology, cat# 7074, RRID:AB_2099233) at 1:1000 or HRP-conjugated horse anti-mouse (Cell Signaling Technology, cat# 7076, RRID:AB_330924) at 1:1000. Quantification was performed using ImageJ software (RRID:SCR_003070), version 1.51 p (*Schneider et al., 2012*). RPMI-7951, A375, and HT-29 cells were also attempted to be probed with two rabbit anti-MAP3K8 antibodies (Signalway Antibody, cat# 33235, RRID:AB_2893080 and Thermo Fisher Scientific, cat# 701352, clone 3H18L5, RRID:AB_2532473), but many nonspecific bands were observed.

## Partial replication: senescence surveillance of premalignant hepatocytes limits liver cancer development

A detailed description of all protocols can be found in the Registered Report (*Raouf et al., 2015*) with detailed protocol information for attempted experiments available on OSF (https://osf.io/82nfe/). A summary of methodological details of results reported are described below.

All animal procedures were approved by the University of California, Davis IACUC and were in accordance with the University of California, Davis policies on the care, welfare, and treatment of laboratory animals. Four- to six-week-old female C.B-17 wild-type (Charles River, strain: C.BKa-*Igh*$^b$/lcrCrl, strain code: 251, RRID:IMSR_CRL:251), SCID/beige (Charles River, strain: CB17.Cg-*Prkdc*$^{scid}$-*Lyst*$^{bg}$/Crl, strain code: 250, RRID:IMSR_CRL:250), BL/6 wild-type (The Jackson Laboratory, strain: C57BL/6J, strain code: 000664, RRID:IMSR_JAX:000664), or CD4$^{-/-}$ (The Jackson Laboratory, strain: B6.129S2-*Cd4*$^{tm1Mak}$/J, strain code: 002663, RRID:IMSR_JAX:002663) mice were housed (2–5 per cage) in ventilated cages that are specific pathogen free, 12 hr light/dark cycles, and fed sterile rodent chow and acidified water ad libitum. Mice were housed for approximately 1 week before being enrolled in the study. The individual mouse was considered the experimental unit within the studies and inclusion/exclusion criteria (e.g., resistance during tail vein injection, or injections taking >10 s) were described in the Registered Report. Housing and experimentation (e.g., injections) were conducted in the same facility.

A pilot study was performed on seven mice. Due to difficulty in the smoothness of the injections with the plasmid solution compared to sodium chloride there was concern that the DNA was not completely dissolved in solutions and forming microprecipitants. Additionally, the anesthetic cocktail appeared to be too strong as mice did not recover as quickly as expected. Following confirmation by the original authors we switched the plasmid maxiprep kit from the kit specified in the Registered Report to another source (Qiagen, cat# 12362) and did not use anesthesia during the injections.

Following the pilot study, a total of 38 C.B-17 wild-type, 30 SCID/beige, 14 BL/6 wild-type, and 26 CD4$^{-/-}$ mice were randomized (stratified into strain-based groups and cohoused so that mice of the same strain receiving the same injections were housed together) to receive the pPGK-SB13 transposase vector and pT/Caggs*Nras*$^{G12V}$ or pT/Caggs*Nras*$^{G12V/D38A}$ transposon vector, which were used in the original study and shared by Dr. Lars Zender (University of Tuebingen), by hydrodynamic tail vein injection as described in the Registered Report. Injections were conducted blinded to the treatment (genotype and vectors) and occurred on nine separate days with between 3 and 14 mice injected on any given day. The following number of mice was excluded (C.B-17 wild-type injected with *Nras*$^{G12V}$ = 5, C.B-17 wild-type injected with *Nras*$^{G12V/D38A}$ = 12, SCID/beige injected with *Nras*$^{G12V}$ = 5, SCID/beige injected with *Nras*$^{G12V/D38A}$ = 4, BL/6 wild-type injected with *Nras*$^{G12V}$ = 2, BL/6 wild-type injected with *Nras*$^{G12V/D38A}$ = 2, CD4$^{-/-}$ injected with *Nras*$^{G12V}$ = 11, CD4$^{-/-}$ injected with *Nras*$^{G12V/D38A}$ = 5) because of unsuccessful injections (e.g., resistance) or death (n = 3). Mice were monitored and euthanized at 6, 12, or 30 days postinjection to harvest liver tissue, which were cleaned with phosphate-buffered aline (PBS) before fixation in 4% paraformaldehyde at 4°C. Samples were shipped at 4°C for immunohistochemistry and imaging and processed within 2 days of harvest. Immunohistochemistry was performed blinded to the sample condition (protocol: https://osf.io/4zsku/) with the following antibodies and

stains: mouse anti-p16 (abcam, cat# ab54210, clone 2D9A12, RRID:AB_881819) at 1:50 dilution, mouse anti-p21 (BD Biosciences, cat# 556431, clone SXM30, RRID:AB_396415) at 1:50 dilution, mouse anti-Nras (Santa Cruz Biotechnology, cat# sc-31, clone F155, RRID:AB_628041) at 1:100 dilution, mouse IgG1 isotype control (Sigma-Aldrich, cat# M5284, clone MOPC21, RRID:AB_1163685) at 1:50 dilution, biotinylated goat anti-mouse (Thermo Fisher Scientific, cat# TM-060-BN, RRID:AB_716945), strepta-vidin–HRP conjugate (Sigma-Aldrich, cat# RPN1231VS), 3,3' diaminobenzidine (DAB) substrate kit (Abcam, cat# ab64238), and Mayer's hematoxylin (Electron Microscopy Sciences, cat# 26043-05). Five random fields from two stained liver sections from each mouse liver and staining condition were blindly evaluated to count the number of positive cells using CaseViewer (3DHISTECH, RRID:SCR_017654), version 2.2.

## Partial replication: IDH mutation impairs histone demethylation and results in a block to cell differentiation

A detailed description of all protocols can be found in the Registered Report (*Richarson et al., 2016*) with detailed protocol information for attempted experiments available on OSF (https://osf.io/vfsbo/). A summary of methodological details of results reported are described below.

HEK293T cells (ATCC, cat# CRL-3216, RRID:CVCL_0063) and 3T3-L1 cells (ATCC, cat# CL-173, RRID:CVCL_0123) were grown in DMEM supplemented with 10 % FBS and 1 % penicillin/strepto-mycin at 37°C in a humidified atmosphere at 5% $CO_2$. Cells were confirmed to be free of myco-plasma contamination as well as confirmed to be the indicated cells by STR DNA profiling using tests performed by IDEXX (Columbia, MO).

HEK293T cells were seeded in 10 cm culture dishes 1 day before transfection. On transfection day, pLPC empty vector, pLPC-IDH1 wild-type, pLPC-IDH1$^{R132H}$, pLPC-IDH2 wild-type, or pLPC-IDH2$^{R172K}$ were transfected with XtremeGENE HP DNA plasmid transfection reagent following the manufactur-er's instructions. Three days after transfection, cells were harvested and divided into two portions: one was lysed in 100 µl RIPA buffer to assess expression level of IDH1 and IDH2; the other portion was lysed to assess histone methylation status. For histone extraction cells were lysed in hypotonic lysis buffer and rotated at 4 °C overnight with $H_2SO_4$ (final concentration 0.2 N), then histones were extracted by TCA to assess expression level of different histone methylation for one biological repeat, lysed in hypotonic lysis buffer and rotated at 4 °C overnight with $H_2SO_4$ (final 0.2 N), then buffer was replaced by 1× Tris buffer for three biological repeats, or histones were extracted by histone extraction kit (Abcam, cat# ab113476) according to the manufacturer's instructions for two biological repeats. Protein concentrations were quantified by BCA and ~10 µg protein was loaded on 4–12% Bis-Tris protein gel, running at 100 V for 30 min and then 160 V for another 1 hr. Gels were transferred to poly-vinylidene fluoride (PDVF) membrane at 340 mA for 40 min, blocked by 5 % nonfat dry milk in TBST and primary antibodies were diluted in 1 % nonfat dry milk in TBST according to the manufacturer's suggestions: rabbit anti-H3 (Cell Signaling Technology, cat# 4499, clone D1H2, RRID:AB_10544537), at 1:1000 dilution, rabbit anti-H3K4me3 (Millipore, cat# 17-614, RRID:AB_11212770) at 1:2000 dilution, rabbit anti-H3K36me3 (Abcam, cat# ab9050, RRID:AB_306966) at 1 µg/ml dilution, rabbit anti-H3K79me2 (Cell Signaling Technology, cat# 9757, RRID:AB_2118448) at 1:1000 dilution, rabbit anti-H3K9me2 (Cell Signaling Technology, cat# 9753, RRID:AB_659848) at 1:1000 dilution, rabbit anti-H3K9me3 (Abcam, cat# ab8898, RRID:AB_306848) at 1 µg/ml dilution, rabbit anti-IDH1 (Proteintech, cat# 12332-1-AP, RRID:AB_2123159) at 1:1500 dilution, or mouse anti-IDH2 (Abcam, cat# ab55271, clone 5F11, RRID:AB_943793) at 1 µg/ml dilution. After primary antibodies, membranes were incu-bated in HRP-conjugated secondary antibodies in 1 % milk in TBST and signals were detected using ECL according to the manufacturer's instructions for one biological repeat, while the other repeats were incubated in IRDye 680RD anti-rabbit or IRDye 800CW anti-mouse at manufacturer's suggested dilution ratio in 1 % nonfat dry milk in TBST. Membrane images and band intensities were quantified with Odyssey Clx Infrared Imaging System and Image Studio Software (LI-COR Biosciences).

To generate retrovirus, HEK293T cells were transfected with pLPC vector, pLPC-IDH2 wild-type, or pLPC-IDH2$^{R172K}$ (1000 ng, respectively) and helper plasmid mix (700 ng pCMVdR8.74 and 350 ng pMDVSVG) per well in 6-well plates using XtremeGene HP DNA Plasmid Transfection Reagent (Roche, cat# 06366244001) (4 µl reagent in 400 µl OPTI-MEM per well) following the manufacturer's instruc-tions. Twenty-four hours after transfection, medium was replaced. After 48 hr post-transfection, super-natant was collected for each sample and centrifuged at 500 × *g* for 10 min at room temperature.

Supernatant was then filtered through a 0.45 µm syringe filter, aliquoted and stored at −80 °C. 3T3-L1 cells per seeded at 50,000 cells/well in 6-well plate 1 day before transduction. On transduction day, medium was replaced with 1.7 ml fresh medium, 300 µl retrovirus of each sample, and polybrene at final concentration of 8 µg/ml. Then spinoculated by spinning at 1000 × *g* for 1 hr at room temperature and then incubated. Two days after transduction, viral transduction medium was replaced by fresh medium with 2.5 µg/ml puromycin, cells were incubated in puromycin for 7 days and then split into biological repeats.

Metabolites were extracted from cells after washing with ice-cold PBS by adding ice-cold 80 % methanol containing 20 µM L-norvaline and incubating for 20 min (1.0 ml for a 10 cm dish). Plates were thawed at room temperature and cells were scraped and extracts transferred to microcentrifuge tubes. Chloroform was added (0.5 ml) and samples were spun at 14,000 × *g* for 20 min at 4 °C. The upper phase was collected and samples were dried using a MiVac. Dried extracts were redissolved in a 1:1 mixture of acetonitrile and *N*-methyl-*N*-tert-butyl dimethyl silyl trifluoroacetamide (MTBSTFA) (60 µl). Samples were heated for 75 min at 75 °C. GC–MS analysis was conducted as described in the Registered Report. In parallel to the samples, a standard curve of known amounts of 2HG, L-glutamate, and L-norvaline were dried, derivatized, and run. 2HG *m/z* 433 and glutamate *m/z* 432 peaks were integrated using A GCMS-QP2010 Plus (Shimadzu Scientific Instruments), and the 2HG/glutamate peak area ratio was calculated. The metabolomics data are available at the NIH Common Fund's Data Repository and Coordinating Center (supported by NIH grant, U01-DK097430) website (http://www.metabolomicsworkbench.org), where it has been assigned a Metabolomics Workbench Project ID: ST000904. The data can be accessed directly via its Project DOI: http://dx.doi.org/10.21228/M8H10W.

## Statistical analysis

Statistical analysis and figure generation were performed with R software (RRID:SCR_001905), version 4.1.0 (*R Development Core Team, 2021*). All data, csv files, and analysis scripts are available on the OSF for each attempted replication (see figure legends for links). Confirmatory statistical analyses were preregistered before the experimental work began as outlined in the Registered Reports. Data were checked to ensure assumptions of statistical tests were met. When described in the results, the Bonferroni correction, to account for multiple testings, was applied to the alpha error or the *p* value. The Bonferroni corrected value was determined by dividing the uncorrected value (0.05) by the number of tests performed. Although the Bonferroni method is conservative, it was accounted for in the power calculations to ensure sample size was sufficient. A meta-analysis of a common original and replication effect size was performed with a random-effects model and the *metafor* R package (RRID:SCR_003450), version 3.0-2 (*Viechtbauer, 2010*). The original study data were extracted a priori from the published figures by estimating the value reported or shared by the original authors. The original summary data were published in the Registered Reports and were used in the power calculations to determine the sample size for this study.

## Acknowledgements

The Reproducibility Project: Cancer Biology would like to thank the following companies for generously donating reagents to the Reproducibility Project: Cancer Biology; American Type and Tissue Collection (ATCC), Applied Biological Materials, BioLegend, Charles River Laboratories, Corning Incorporated, DDC Medical, EMD Millipore, Harlan Laboratories, LI-COR Biosciences, Mirus Bio, Novus Biologicals, Sigma-Aldrich, and System Biosciences (SBI). We thank Crystal Newell, Jacqueline Carrell, and Laura Skinner for help with literature review.

## Additional information

### Competing interests

Timothy M Errington: Employed by the nonprofit Center for Open Science that has a mission to increase openness, integrity, and reproducibility of research. Alexandria Denis: Was employed by the nonprofit Center for Open Science that has a mission to increase openness, integrity, and

reproducibility of research. Renee Araiza, Lynette R Bower, Jessica Campos, Sarah Denson, Todd Tolentino, Brandon Willis, Joshua Wood: Employed by the University of California, Davis Mouse Biology Program, a Science Exchange associated lab during experimentation. Pedro Aza-Blanc: Was employed by Cancer Metabolism Facility at Sanford Burnham Prebys Medical Discovery Institute, a Science Exchange associated lab during experimentation. Heidi Chu, Jennie Kwok, Vidhu Sharma, Angela Trinh, Lisa Young: Employed by Applied Biological Materials, a Science Exchange associated lab during experimentation. Cristine Donham, Babette Haven, Michael Settles: Was employed by TGA Sciences, a Science Exchange associated lab during experimentation. Elizabeth Iorns: Employed by and hold shares in Science Exchange Inc. Elysia McDonald, Steven Pelech: Nicole Perfito, Rachel Tsui: Was employed by and hold shares in Science Exchange Inc. Amanda Pike: Was employed by Applied Biological Materials, a Science Exchange associated lab during experimentation. Darryl Sampey: Employed by BioFactura, a Science Exchange associated lab during experimentation. David A Scott: Employed by Cancer Metabolism Facility at Sanford Burnham Prebys Medical Discovery Institute, a Science Exchange associated lab during experimentation. The other authors declare that no competing interests exist.

### Funding

The Reproducibility Project: Cancer Biology was supported by a grant from Arnold Ventures (formerly known as the Laura and John Arnold Foundation) provided to the Center for Open Science in collaboration with Science Exchange. The funder had no role in study design, data collection and interpretation, or the decision to submit the work for publication.

### Author contributions

Timothy M Errington, Conceptualization, Data curation, Formal analysis, Project administration, Supervision, Validation, Visualization, Writing – original draft, Writing – review and editing; Alexandria Denis, Data curation, Formal analysis, Project administration, Validation, Visualization; Anne B Allison, Writing – original draft, Writing – review and editing; Renee Araiza, Lynette R Bower, Jessica Campos, Sarah Denson, Todd Tolentino, Brandon Willis, Joshua Wood, Data curation, Investigation, Resources (Kang et al., 2012 replication attempt); Pedro Aza-Blanc, David A Scott, Data curation, Investigation, Resources (Lu et al., 2012 replication attempt); Heidi Chu, Jennie Kwok, Amanda Pike, Vidhu Sharma, Angela Trinh, Lisa Young, Data curation, Investigation, Resources (Kan et al., 2010 and Johannessen et al., 2010 replication attempts); Cristine Donham, Babette Haven, Elysia McDonald, Michael Settles, Data curation, Investigation, Resources (Sharma et al., 2010 replication attempt); Kaitlyn Harr, Writing – original draft; Elizabeth Iorns, Project administration, Supervision; Steven Pelech, Writing – review and editing; Nicole Perfito, Rachel Tsui, Project administration; Darryl Sampey, Data curation, Investigation, Resources (Ricci-Vitiani et al., 2010 replication attempt)

### Author ORCIDs

Timothy M Errington http://orcid.org/0000-0002-4959-5143
Anne B Allison http://orcid.org/0000-0002-6941-3159
Cristine Donham http://orcid.org/0000-0002-5590-7502
Elizabeth Iorns http://orcid.org/0000-0002-5515-1258
David A Scott http://orcid.org/0000-0002-8668-2449

### Ethics

All animal procedures were approved by the University of California, Davis IACUC and were in accordance with the University of California, Davis policies on the care, welfare, and treatment of laboratory animals.

### Decision letter and Author response

Decision letter https://doi.org/10.7554/eLife.73430.sa1
Author response https://doi.org/10.7554/eLife.73430.sa2

## Additional files

### Supplementary files

• Transparent reporting form

## Data availability

All experimental details (e.g., additional protocol details, data, analysis files) of the individual replications and data, code, and materials for the overall project are openly available at https://osf.io/collections/rpcb/ (see figure legends for links to individual studies). The metabolomics data are available at the NIH Common Fund's Data Repository and Coordinating Center (supported by NIH grant, U01-DK097430) website, (http://www.metabolomicsworkbench.org), where it has been assigned a Metabolomics Workbench Project ID: ST000904. The data can be accessed directly via it's Project DOI: https://doi.org/10.21228/M8H10W.

The following dataset was generated:

| Author(s) | Year | Dataset title | Dataset URL | Database and Identifier |
|---|---|---|---|---|
| Donham C, Haven B, McDonald E, Settles M, Iorns E, Tsui R, Denis A, Perfito N, Errington TM | 2021 | Study 2: Replication of Sharma et al., 2010 (Cell) | http://dx.doi.org/10.17605/OSF.IO/XBIGN | Open Science Framework, 10.17605/OSF.IO/XBIGN |
| Courville P, Sampey D, Iorns E, Tsui T, Denis A, Perfito N, Errington TM | 2021 | Study 5: Replication of Ricci-Vitiani et al., 2010 (Nature) | http://dx.doi.org/10.17605/OSF.IO/MPYVX | Open Science Framework, 10.17605/OSF.IO/MPYVX |
| Chu H, Kwok J, Pike A, Sharma V, Trinh A, Young L, Iorns E, Tsui R, Denis A, Perfito N, Errington TM | 2021 | Study 6: Replication of Kan et al., 2010 (Nature) | http://dx.doi.org/10.17605/OSF.IO/JPEQG | Open Science Framework, 10.17605/OSF.IO/JPEQG |
| Bhargava A, Iorns E, Tsui R, Denis A, Perfito N, Errington TM | 2021 | Study 7: Replication of Heidorn et al., 2010 (Cell) | http://dx.doi.org/10.17605/OSF.IO/B1AW6 | Open Science Framework, 10.17605/OSF.IO/B1AW6 |
| Chu H, Kwok J, Pike A, Sharma V, Trinh A, Young L, Iorns E, Tsui R, Denis A, Perfito N, Errington TM | 2021 | Study 12: Replication of Johannessen et al., 2010 (Nature) | http://dx.doi.org/10.17605/OSF.IO/LMHJG | Open Science Framework, 10.17605/OSF.IO/LMHJG |
| Araiza R, Bower LR, Campos J, Denson S, Raouf S, Tolentino T, Weston C, Willis B, Wood J, Iorns E, Tsui R, Denis A, Perfito N, Errington TM | 2021 | Study 34: Replication of Kang et al., 2011 (Nature) | http://dx.doi.org/10.17605/OSF.IO/82NFE | Open Science Framework, 10.17605/OSF.IO/82NFE |
| Aza-Blanc P, Scott DA, Iorns E, Tsui R, Denis A, Perfito N, Errington TM | 2021 | Study 47: Replication of Lu et al., 2012. (Nature) | http://dx.doi.org/10.17605/OSF.IO/VFSBO | Open Science Framework, 10.17605/OSF.IO/VFSBO |
| Scott DA | 2018 | Measurement of 2-hydroxyglutarate for reproducibility project | http://dx.doi.org/10.21228/M8H10W | Metabolomics Workbench, 10.21228/M8H10W |

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
