## [Editor Report]

This article provides a succinct presentation of the remaining unfinished Registered Reports for the Reproducibility Project: Cancer Biology. The article will be useful for evaluating the success of the reproducibility project.

---

## [Decision Letter]

**Decision letter after peer review:**

Thank you for submitting your work entitled "Results of partially completed replications from the Reproducibility Project: Cancer Biology" for further consideration by *eLife*. Your article has has been reviewed by 2 peer reviewers, one of whom is a member of our Board of Reviewing Editors, and the evaluation has been overseen by a Senior Editor. The reviewers have opted to remain anonymous.

As is customary in *eLife*, the reviewers have discussed their critiques with one another. What follows below is the Reviewing Editor's edited compilation of the essential and ancillary points provided by reviewers in their critiques and in their interaction post-review. Please submit a revised version that addresses these concerns directly. Although we expect that you will address these comments in your response letter, we also need to see the corresponding revision clearly marked in the text of the manuscript. Some of the reviewers' comments may seem to be simple queries or challenges that do not prompt revisions to the text. Please keep in mind, however, that readers may have the same perspective as the reviewers. Therefore, it is essential that you attempt to amend or expand the text to clarify the narrative accordingly.

Reviewers' Comments:

This is a summary of the results obtained from incomplete replication reports from the Cancer Reproducibility Project. This manuscript is a sweeping but inelegantly executed summary of incomplete replication studies. The data are a collection of experiments covering diverse discoveries reported by many investigators. The authors describe their replication experiments in the context of the original research data and simplistically conclude whether the current work agrees with, or is in opposition to, the original research. But all too often, the reports lack a critical evaluation of why the replication experiments may have failed to repeat the original research. This uncritical examination may leave the reader to unfairly doubt the previously published research, which, in the absence of a complete evaluation would be utterly unfair. While a strong argument can be made for publishing negative and contrary results, incomplete and contradictory results uncritically evaluated are of little value.

1) The Abstract is vague. While the research is sweeping and difficult to adequately summarize, the abstract must give the reader a sense of the scope, a summary of the results and a succinct and compelling conclusion that highlights the value of the effort.

2) The conclusions section is a simplistic summary of extensive efforts to reproduce important research and the difficulties of replicating them. No conclusions may be drawn as to why, but the many possible explanations, including limitations in technical expertise of the authors is not adequately addressed. If the authors are not going to provide detailed explanations in the discussion regarding why they might not have been able to replicate the studies, they should do so at the end of the Results section for each study. The lesson learned from this effort may be that the expertise required to repeat detailed and technically complex research is not straightforward even with a detailed methods and reagents list. But that should in no way be implied by the authors as a weakness in the original research. Stating any less than this unfairly detrimental to the original research.

3) Line 28 "… attempts were completed and ultimately stopped so efforts could be focused on completing as many replications as possible." This statement is confusing. Are the authors stating that prioritized some studies over others?

4) Lines 34-36. "While the unfinished experiments cannot address whether, under the conditions of the original study, the original findings were observed, they are still results that can advance our methodological understanding." This statement is vague and not compelling.

Partial Replication: A chromatin-mediated reversible drug-tolerant state in cancer 62 cell subpopulations

5) A concern here is that the differences found don't impact the key result. The PC9 cells these investigators used were less resistant than those of the original authors. That's not a big deal if the key biological principle had been tested. As written, this criticizes the original paper on a point that may not be substantive.

6) Lines 90-92. "…which both appeared to require higher concentrations of erlotinib to achieve a half-maximal inhibitory concentration compared to what was reported in the original study." The authors should be more specific in the text.

Partial Replication: Diverse somatic mutation patterns and pathway alterations in 191 human cancers.

7) Line 221. "Led", instead of "lead"?

8) Lines 230-231. This suggests that the NIH3T3 cells used in this attempt acquired the ability to survive and proliferate in soft agar without attachment." This mistaken result can occur when soft agar is not poured properly. The cells grow on the dish surface under the agar. Are the authors certain the cells grew in, and not below, the agar? It is highly unlikely that NIH 3T3 cells from such a carefully controlled experiment all spontaneously transformed. Moreover, the authors should confirm the provenance of the NIH3T3 cells – it is stated they were obtained from ATCC, but what is the history of the cells between arrival from ATCC and their use in this experiment.

Partial Replication: Kinase-dead BRAF and oncogenic RAS cooperate to drive tumor progression through CRAF

9) Line 304-305. "However, due to resource constraints…" This is vague and should be more thoroughly explained.

Partial Replication: RAF inhibitors prime wild-type RAF to activate the MAPK pathway and enhance growth.

10) There is serious concern that the authors are subverting the previous review process of this Replication Study by publishing the data here. The original replication report was rejected from *eLife* because of the low technical quality of the studies presented and because of the inclusion of studies not described in the Registered Report. Both of these problems are present in this section of the manuscript. We recommend that the first paragraph is retained (except for deletion of the last sentence) and that all the remaining text and figures are deleted. The citation to the pre-print at the end of the first paragraph is sufficient to notify interested readers about the study that was performed.

Partial Replication: COT drives resistance to RAF inhibition through MAP kinase pathway reactivation.

11) There is no citation to Figure 6B or S6B. This should be added to the paragraph (lines 560-566).

12) Figure 6B. The Y axis labels are pMEK/ERK and pERK/MEK. This is OK, but this is not what was done by the original authors. Is there a reason not use pERK/ERK and pMEK/MEK?

13) Line 617. "supporting" instead of "supported"?

14) Lines 629-630 "628 Surprisingly, the Nras expression for both experiments were quite low, with many at, or 629 near, zero percent confounding interpretation of these data." How does the reader interpret this statement? Is this a technical issue unique to this replication study?

15) Lines 696-697. "… due to s lack of differentiation induced in control conditions, despite a successful pilot assay."

Partial Replication: IDH mutation impairs histone demethylation and results in a block to cell differentiation.

15) Lines 702-704. "…these data are unable to address whether, under the conditions of the original study, mutant IDH2 is correlated with a block in differentiation and if this effect was mediated by the histone demethylase KDM4Cs." The authors should indicate that their inability to replicate the results were possibly due to their own technical limitations.

16) Figure 8D, S8B and S8D to be illegible.

---

## [Author Response]

Reviewers' Comments:This is a summary of the results obtained from incomplete replication reports from the Cancer Reproducibility Project. This manuscript is a sweeping but inelegantly executed summary of incomplete replication studies. The data are a collection of experiments covering diverse discoveries reported by many investigators. The authors describe their replication experiments in the context of the original research data and simplistically conclude whether the current work agrees with, or is in opposition to, the original research. But all too often, the reports lack a critical evaluation of why the replication experiments may have failed to repeat the original research. This uncritical examination may leave the reader to unfairly doubt the previously published research, which, in the absence of a complete evaluation would be utterly unfair. While a strong argument can be made for publishing negative and contrary results, incomplete and contradictory results uncritically evaluated are of little value.1) The Abstract is vague. While the research is sweeping and difficult to adequately summarize, the abstract must give the reader a sense of the scope, a summary of the results and a succinct and compelling conclusion that highlights the value of the effort.

We have substantially revised the abstract to address these concerns.

2) The conclusions section is a simplistic summary of extensive efforts to reproduce important research and the difficulties of replicating them. No conclusions may be drawn as to why, but the many possible explanations, including limitations in technical expertise of the authors is not adequately addressed. If the authors are not going to provide detailed explanations in the discussion regarding why they might not have been able to replicate the studies, they should do so at the end of the Results section for each study. The lesson learned from this effort may be that the expertise required to repeat detailed and technically complex research is not straightforward even with a detailed methods and reagents list. But that should in no way be implied by the authors as a weakness in the original research. Stating any less than this unfairly detrimental to the original research.

We have revised the manuscript to include further discussion in sections of each study where relevant and an extended conclusion section. However, we disagree with the reviewers that this implies that expertise is required – a more likely scenario is the lack of detailed methods and reagents lists despite attempts to gather these from the original papers and authors. The challenges in attempting to conduct these experiments, as well as all experiments attempted during the Reproducibility Project: Cancer Biology, are a focus of a complimentary summary paper. We include reference to this paper and the other summary paper of the outcomes of completed experiments. Finally, we agree with the reviewers that we can not draw conclusions about why we were unable to complete some of these experiments since there are many possible explanations which were not explored. Instead we conclude that we stopped these experiments due to mundane technical or unanticipated methodological challenges as we navigated attempting to complete as many experiments during the course of the Reproducibility Project: Cancer Biology.

3) Line 28 "… attempts were completed and ultimately stopped so efforts could be focused on completing as many replications as possible." This statement is confusing. Are the authors stating that prioritized some studies over others?

We substantially revised the abstract to address this concern and the concern (#1) raised above. We also included a sentence in the revised introduction to elaborate why some experiments were stopped.

4) Lines 34-36. "While the unfinished experiments cannot address whether, under the conditions of the original study, the original findings were observed, they are still results that can advance our methodological understanding." This statement is vague and not compelling.

We substantially revised the abstract to address this concern and the concerns (#1 and 3) raised above.

Partial Replication: A chromatin-mediated reversible drug-tolerant state in cancer 62 cell subpopulations5) A concern here is that the differences found don't impact the key result. The PC9 cells these investigators used were less resistant than those of the original authors. That's not a big deal if the key biological principle had been tested. As written, this criticizes the original paper on a point that may not be substantive.

We agree that this might not impact the key biological principle tested, however the reason for the difference is unknown. For example, is it the PC9 cells as the reviewers suggest or the drug, or a combination of both? Importantly this is not a criticism of the original paper, rather it is what we observed during the replication that was unexpected and raises questions for the replication moving forward, such as do the results of downstream experiments differ when experiments are performed using the same dose as the original study vs the higher concentration suggested in these preliminary results – or is a different set of conditions necessary (e.g., is it necessary to achieve ~0.3% cell survival after 9 days?). While we did not further pursue these questions due to stopping this experiment, the results of the experiments proposed in the Registered Report under these different conditions would help provide insight into what conditions are necessary to obtain the originally reported results. We have revised the end of this section to highlight these considerations and to minimize the potential of this being perceived as a criticism of the original paper.

6) Lines 90-92. "…which both appeared to require higher concentrations of erlotinib to achieve a half-maximal inhibitory concentration compared to what was reported in the original study." The authors should be more specific in the text.

We included the estimated half-maximal inhibitory concentrations of the replication attempt (survival: ~0.15 µM; EGFR: >1 µM) and the original study (survival: ~0.0085 µM; EGFR: <0.01 µM) (original paper estimated from Figures 1E and 2E)

Partial Replication: Diverse somatic mutation patterns and pathway alterations in 191 human cancers.7) Line 221. "Led", instead of "lead"?

Fixed.

8) Lines 230-231. This suggests that the NIH3T3 cells used in this attempt acquired the ability to survive and proliferate in soft agar without attachment." This mistaken result can occur when soft agar is not poured properly. The cells grow on the dish surface under the agar. Are the authors certain the cells grew in, and not below, the agar? It is highly unlikely that NIH 3T3 cells from such a carefully controlled experiment all spontaneously transformed. Moreover, the authors should confirm the provenance of the NIH3T3 cells – it is stated they were obtained from ATCC, but what is the history of the cells between arrival from ATCC and their use in this experiment.

We agree this is a possible scenario and one we cannot rule out. We have removed this section from the manuscript to avoid confusion of these results. Of note, we did confirm the soft agar assay was performed to enable colony formation with the HMEC cells, although variation in cell lines and/or plate preparation could have resulted in this outcome. Also, the NIH3T3 cells were sent right before the start of the experiment, so they were handled following ATCC recommendations and propagated briefly before conducting the experiments (i.e., the cells were not passaged or stored for extended periods of time between expanding of the cells).

Partial Replication: Kinase-dead BRAF and oncogenic RAS cooperate to drive tumor progression through CRAF9) Line 304-305. "However, due to resource constraints…" This is vague and should be more thoroughly explained.

We revised this sentence to better reflect the reason. We also included a sentence in the revised introduction to elaborate why some experiments were stopped.

Partial Replication: RAF inhibitors prime wild-type RAF to activate the MAPK pathway and enhance growth.10) There is serious concern that the authors are subverting the previous review process of this Replication Study by publishing the data here. The original replication report was rejected from eLife because of the low technical quality of the studies presented and because of the inclusion of studies not described in the Registered Report. Both of these problems are present in this section of the manuscript. We recommend that the first paragraph is retained (except for deletion of the last sentence) and that all the remaining text and figures are deleted. The citation to the pre-print at the end of the first paragraph is sufficient to notify interested readers about the study that was performed.

We appreciate this response and were not attempting to subvert the previous review process for this Replication Study. Rather we saw this paper as a way to present the experimental results from Registered Reports that were not submitted or not accepted as Replication Studies. However, we appreciate this perspective and agree that removing the figures, text, etc is best, especially since we’re planning to post this as a preprint. We have revised the manuscript accordingly.

Partial Replication: COT drives resistance to RAF inhibition through MAP kinase pathway reactivation.11) There is no citation to Figure 6B or S6B. This should be added to the paragraph (lines 560-566)

Fixed.

12) Figure 6B. The Y axis labels are pMEK/ERK and pERK/MEK. This is OK, but this is not what was done by the original authors. Is there a reason not use pERK/ERK and pMEK/MEK?

This was a recommendation during peer review of the Registered Report. We have added this in the manuscript to explain this.

13) Line 617. "supporting" instead of "supported"?

Fixed.

14) Lines 629-630 "628 Surprisingly, the Nras expression for both experiments were quite low, with many at, or 629 near, zero percent confounding interpretation of these data." How does the reader interpret this statement? Is this a technical issue unique to this replication study?

We have revised this section to include a more thorough discussion of what this unexpected outcome means.

15) Lines 696-697. "… due to s lack of differentiation induced in control conditions, despite a successful pilot assay."

We elaborated this sentence to explain the context of the pilot assay.

Partial Replication: IDH mutation impairs histone demethylation and results in a block to cell differentiation.15) Lines 702-704. "…these data are unable to address whether, under the conditions of the original study, mutant IDH2 is correlated with a block in differentiation and if this effect was mediated by the histone demethylase KDM4Cs." The authors should indicate that their inability to replicate the results were possibly due to their own technical limitations.

We removed the ‘under the conditions of the original study’ here and in other locations mentioned. We agree that technical limitations is one possible reason for why the results could not be replicated, but so are many other reasons from an error or lack of detail in the original study to variations in biological systems. We have elaborated on this aspect in the conclusion section.

16) Figure 8D, S8B and S8D to be illegible.

Fixed.